# A dual function of the IDA peptide in regulating cell separation and modulating plant immunity at the molecular level

Vilde Olsson Lalun[1], Maike Breiden[2†], Sergio Galindo-Trigo[1†], Elwira Smakowska-Luzan[3], Rüdiger GW Simon[2], Melinka A Butenko[1]*

[1]Section for Genetics and Evolutionary Biology, Department of Biosciences, University of Oslo, Oslo, Norway; [2]Institute for Developmental Genetics and Cluster of Excellence on Plant Sciences, Heinrich Heine University, Düsseldorf, Germany; [3]Gregor Mendel Institute (GMI), Austrian Academy of Sciences, Vienna Biocenter (VBC), Vienna, Austria

**Abstract** The abscission of floral organs and emergence of lateral roots in *Arabidopsis* is regulated by the peptide ligand inflorescence deficient in abscission (IDA) and the receptor protein kinases HAESA (HAE) and HAESA-like 2 (HSL2). During these cell separation processes, the plant induces defense-associated genes to protect against pathogen invasion. However, the molecular coordination between abscission and immunity has not been thoroughly explored. Here, we show that IDA induces a release of cytosolic calcium ions ($Ca^{2+}$) and apoplastic production of reactive oxygen species, which are signatures of early defense responses. In addition, we find that IDA promotes late defense responses by the transcriptional upregulation of genes known to be involved in immunity. When comparing the IDA induced early immune responses to known immune responses, such as those elicited by flagellin22 treatment, we observe both similarities and differences. We propose a molecular mechanism by which IDA promotes signatures of an immune response in cells destined for separation to guard them from pathogen attack.

*For correspondence:
m.a.butenko@ibv.uio.no

†These authors contributed equally to this work

Competing interest: The authors declare that no competing interests exist.

## eLife assessment

This manuscript presents **valuable** findings on the role of a plant peptide in coordinating developmental and immune responses signaling. The evidence supporting the claims, while mainly descriptive and somewhat limited due to the main conclusions being drawn from overexpression lines, is mostly **solid**. The findings are interesting, they align with existing models, and they are of relevance to plant pathologists and developmental biologists.

## Introduction

All multicellular organisms require tightly regulated cell-to-cell communication during development and adaptive responses. In addition to hormones, plants use small-secreted peptide ligands to regulate these highly coordinated events of growth, development and responses to abiotic and biotic stress (*Matsubayashi, 2011*; *Olsson et al., 2019*). In plants, organ separation or abscission, involves cell separation between specialized abscission zone (AZ) cells and enables the removal of unwanted or diseased organs in response to endogenous developmental signals or environmental stimuli. *Arabidopsis thaliana* (*Arabidopsis*) abscise floral organs (stamen, petals, and sepals) after pollination, and this system has been used as a model to study the regulation of cell separation and abscission (*Bleecker and Patterson, 1997*). Precise cell separation also occurs during the emergence of new

lateral root (LR) primordia, root cap sloughing, formation of stomata and radicle emergence during germination (*Roberts et al., 2000*). In *Arabidopsis*, the inflorescence deficient in abscission (IDA) peptide-mediated signaling system ensures the correct spatial and temporal abscission of sepals, petals, and stamens. The IDA protein shares a conserved C-terminal domain with the other 7 IDA-like (IDL) members (*Butenko et al., 2003*; *Vie et al., 2015*) which is processed to a functional 12 amino acid hydroxylated peptide (*Butenko et al., 2014*). The abscission process is regulated by the production and release of IDA, which binds the genetically redundant plasma membrane (PM) localized receptor kinases (RKs), HAESA (HAE), and HAESA-like 2 (HSL2). IDA binding to HAE and HSL2 promotes receptor association with members of the co-receptor somatic embryogenesis receptor kinase (SERK) family and further downstream signaling leading to cell separation events. (*Cho et al., 2008*; *Meng et al., 2016*; *Santiago et al., 2016*; *Stenvik et al., 2008*). Recently, it has been shown that IDA and IDL family members also bind and activate HSL1 in the regulation of leaf epidermal pattering, indicating subfunctionalisation within this clade of receptors (*Roman et al., 2022*) and opens for the possibility that HAE or HSL2 could have additional functions than regulating cell separation events.

A deficiency in the IDA signaling pathway prevents the expression of genes encoding secreted cell wall remodeling and hydrolase enzymes thus hindering floral organs to abscise (*Butenko et al., 2003*; *Cho et al., 2008*; *Kumpf et al., 2013*; *Meng et al., 2016*; *Niederhuth et al., 2013*; *Stenvik et al., 2008*; *Taylor and Walker, 2018*). Interestingly, components of the IDA signaling pathway control different cell separation events during *Arabidopsis* development. IDA signaling through HAE and HSL2 regulates cell wall remodeling genes in the endodermal, cortical, and epidermal tissues overlaying the LR primordia during LR emergence (*Kumpf et al., 2013*). In addition, IDL1 signals through HSL2 to regulate cell separation during root cap sloughing (*Shi et al., 2018*).

Plant cell walls act as physical barriers against pathogenic invaders. The cell-wall processing and remodeling that occurs during abscission leads to exposure of previously protected tissue. This provides an entry point for phytopathogens thereby increasing the need for an effective defense system in these cells (*Agustí et al., 2009*). Indeed, it has been shown that AZ cells undergoing cell separation express defense genes and it has been proposed that they function in protecting the AZ cells from infection after abscission has occurred (*Cai and Lashbrook, 2008*; *Niederhuth et al., 2013*). *IDA* elicits the production of reactive oxygen species (ROS) in *Nicotiana benthamiana* (*N. benthamiana*) leaves transiently expressing the HAE or HSL2 receptor (*Butenko et al., 2014*). As a consequence of defense responses, ROS occurs in the apoplast and is involved in regulating the cell wall structure by producing cross-linking of cell wall components in the form of hydrogen peroxide ($H_2O_2$) or act as cell wall loosening agents in the form of hydroxyl radical ($\cdot OH$) radicals (*Kärkönen and Kuchitsu, 2015*). ROS may also serve as a direct defense response by their highly reactive and toxic properties (*Kärkönen and Kuchitsu, 2015*). In addition, ROS may act as a secondary signaling molecule enhancing intracellular defense responses as well as contributing to the induction of defense-related genes (reviewed in *Castro et al., 2021*). In *Arabidopsis*, apoplastic ROS is mainly produced by a family of NADPH oxidases located in the PM known as respiratory burst oxidase proteins (RBOHs; *Torres and Dangl, 2005*). The production of ROS is often found closely linked to an increase in the concentration of cytosolic $Ca^{2+}$ ($[Ca^{2+}]_{cyt}$), where the interplay between these signaling molecules is observed in immunity across kingdoms (*Kadota et al., 2015*; *Steinhorst and Kudla, 2013*). Interestingly, ROS and $[Ca^{2+}]_{cyt}$ are known signaling molecules acting downstream of several peptide ligand-receptor pairs. This includes the endogenous PAMPs elicitor peptide (PEP)1, PEP2, and PEP3 which function as elicitors of systemic responses against pathogen attack and herbivory (*Huffaker et al., 2006*; *Ma et al., 2012*; *Qi et al., 2010*), as well as the pathogen derived peptide ligands flagellin22 (flg22) and elongation factor thermo unstable (Ef-Tu) which bind and activate defense-related RKs in the PM of the plant cell (*Felix et al., 1999*; *Gómez-Gómez et al., 1999*; *Ranf et al., 2011*).

The interplay between ROS and $[Ca^{2+}]_{cyt}$ has been well studied in the flg22-induced signaling system. The flg22 peptide interacts with the extracellular domain of the flagellin sensing 2 (FLS2) receptor kinase, which rapidly leads to an increase in $[Ca^{2+}]_{cyt}$ (*Ranf et al., 2011*). In a resting state, the co-receptor brassinosteroid insensitive1-associated receptor kinase 1/SERK3 (BAK1) and the cytoplasmic kinase botrytis-induced kinase 1 (BIK1) form a complex with FLS2. Upon binding of flg22 to the FLS2 receptor, BIK1 is released from the receptor complex and activates intracellular signaling molecules through phosphorylation, among other, the PM localized RBOHD (*Kadota et al., 2014*;

*Li et al., 2014*). The flg22-induced apoplastic ROS produced binds to PM localized $Ca^{2+}$ channels on neighboring cells, activating $Ca^{2+}$ influx into the cytosol. The cytosolic $Ca^{2+}$ may bind to calcium-dependent protein kinase5 (CDPKs) regulating RBOHD activity through phosphorylation. The $Ca^{2+}$-dependent phosphorylation of RBOH enhances ROS production, which in turn leads to a propagation of the ROS/$Ca^{2+}$ signal through the plant tissue (*Dubiella et al., 2013*).

A rapid increase in $[Ca^{2+}]_{cyt}$ is not only observed in peptide-ligand-induced signaling. Similar $[Ca^{2+}]_{cyt}$ changes are observed as responses to developmental signals, such as extracellular auxin and environmental signals, such as high-salt conditions and drought (reviewed in *Kudla et al., 2018*). Also, $[Ca^{2+}]_{cyt}$ signals are observed in single cells during pollen tube and root hair growth (*Monshausen et al., 2008*; *Sanders et al., 1999*). How cells decipher the $[Ca^{2+}]_{cyt}$ changes into a specific cellular response is still largely unknown, however, the cell contains a large toolkit of $Ca^{2+}$ sensor proteins that may affect cellular function through changes in protein phosphorylation and gene expression. Each $Ca^{2+}$ binding protein has specific affinities for $Ca^{2+}$ allowing them to respond to different $Ca^{2+}$ concentrations providing a specific cellular response (*Geiger et al., 2010*; *Scherzer et al., 2012*).

Given our previous studies showing IDA induced apoplastic ROS production (*Butenko et al., 2014*), and the known link between ROS and $[Ca^{2+}]_{cyt}$ response observed for other ligands such as flg22 (*Dubiella et al., 2013*), we sought to investigate if IDA induces a release of $[Ca^{2+}]_{cyt}$. Here we report that IDA is able to induce an intracellular release of $Ca^{2+}$ in *Arabidopsis* root tips as well as in the AZ. Pursuing the tight connection between ROS and $Ca^{2+}$ in other defense signaling systems, we explored if the the observed production of ROS and $Ca^{2+}$ in the IDA-HAE/HSL2 signaling pathway could be linked to the involvement of IDA in regulating plant defense (*Patharkar et al., 2017*). We show that a range of biotic and abiotic signals can induce *IDA* expression and we demonstrate that the IDA signal modulates responses of plant immune signaling. We propose that, in addition to regulating cell separation, IDA plays a role in enhancing a defense response in tissues undergoing cell separation thus providing the essential need for enhanced defense responses in tissue during the cell separation event.

## Results

### IDA induces apoplastic production of ROS and elevation in cytosolic calcium-ion concentration

We have previously shown that a 12 amino acid functional IDA peptide, hereby referred to as mIDA (*Supplementary file 1*), elicits the production of a ROS burst in *N.benthamiana* leaves transiently expressing *HAE* or *HSL2* (*Butenko et al., 2014*). To investigate whether mIDA could elicit a ROS burst in *Arabidopsis*, a luminol-dependent ROS assay on *hae hsl2* mutant rosette leaves expressing the full length HAE receptor under a constitutive promoter (*35 S:HAE-YFP*) was performed. *hae hsl2 35 S:HAE-YFP* leaf discs treated with mIDA emitted extracellular ROS, whereas no response was observed in wild-type (WT) leaves (*Figure 1—figure supplement 1a–c*). We investigated the activity of the *HAE* and *HSL2* promoter in 22 days old *Arabidopsis* rosette leaves by cloning the promoters fused to the nuclear localized YFP-derived fluorophore Venus protein (*pHAE:Venus-H2B*, *pHSL2:Venus-H2B*) and could observe a decrease in promoter activity in the oldest rosette leaves giving a possible explanation to the lack of mIDA induced ROS in true leaves (*Figure 1—figure supplement 1d and e*). Since rapid production of ROS is often linked to an elevation in $[Ca^{2+}]_{cyt}$ (*Steinhorst and Kudla, 2013*), we aimed to investigate if mIDA could induce a $[Ca^{2+}]_{cyt}$ response. We imaged $Ca^{2+}$ in 10-day-old roots expressing the cytosolic localized fluorescent $Ca^{2+}$ sensor R-GECO1 (*Keinath et al., 2015*). As a positive control for $[Ca^{2+}]_{cyt}$ release we added 1 mM extracellular ATP (eATP), which leads to a $[Ca^{2+}]_{cyt}$ elevation in the root tip within a minute after application (*Figure 1a*; *Breiden et al., 2021*). Following application of 1 µM mIDA, we detected an increase in $[Ca^{2+}]_{cyt}$ (*Figure 1a* and *Figure 1—video 1*). R-GECO1 fluorescence intensities normalized to background intensities (ΔF/F) were measured from a region of interest (ROI) covering the meristematic and elongation zone of the root and revealed that the response starts 4–5 min after application of the mIDA peptide and lasts for 7–8 min (*Figure 1a*). The signal initiated in the root meristematic zone from where it spread toward the elongation zone and root tip, a second wave was observed in the meristematic zone continuing with $Ca^{2+}$ spikes. The signal amplitude was at a maximum within the elongation zone and decreased as the signal spread (*Figure 1a*).

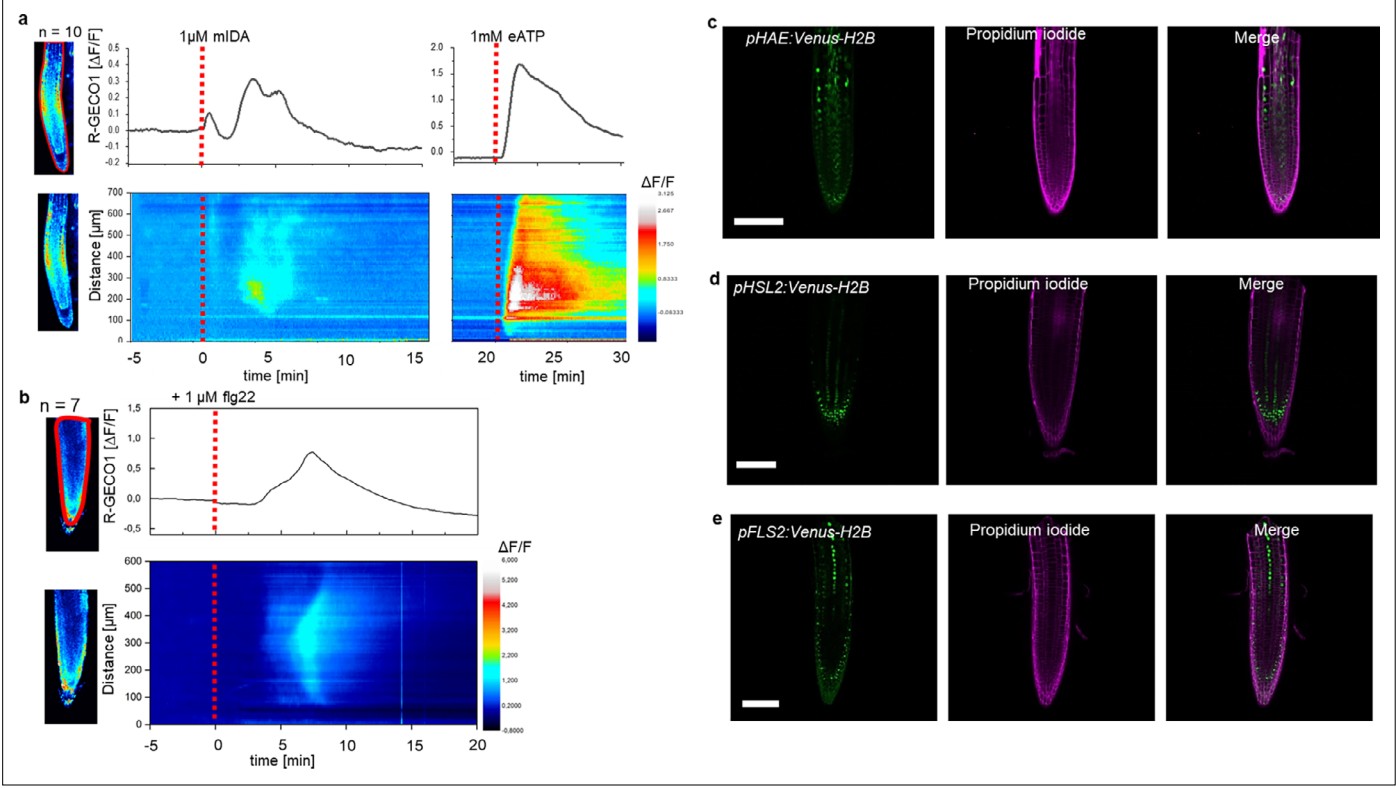

**Figure 1.** mIDA-induced $[Ca^{2+}]_{cyt}$ release in *Arabidopsis* roots correlates with *pHAE* and *pHSL2* activity. (**a**) Normalized R-GECO1 fluorescence intensities (ΔF/F) were measured from regions of interest (ROI) (upper panel, outlined in red) in the meristematic and elongation zone of the root. Fluorescence intensities (ΔF/F) over time of the whole root represented in a heat map (lower panel). Shown are cytosolic calcium concentration ($[Ca^{2+}]_{cyt}$) dynamics in the ROI in response to 1 μM mIDA over time. (see also *Figure 1—video 1*). Red lines at 5 min (min) indicates application of mIDA peptide or application of eATP at 22 min. Representative response from 10 roots (*Figure 1—figure supplement 2*). The increase in $[Ca^{2+}]_{cyt}$ response propagates through the roots as two waves. (**b**) For comparison; Normalized R-GECO1 fluorescence intensities (ΔF/F) measured from regions of interest (ROI) (outlined in red, upper panel) in response to 1 μM flg22 over time. Fluorescence intensities (ΔF/F) over time of the whole root represented in a heat map (lower panel). Red line at 0 min (min) indicates application of flg22 peptide. Representative response from 7 roots (*Figure 1—figure supplement 2*). The increase in $[Ca^{2+}]_{cyt}$ response propagates through the roots as a single wave seen as normalized R-GECO1 fluorescence intensities (ΔF/F) shown as a heat map (see also *Figure 1—video 2*). (**c, d, e**) Expression of the receptors (**c**), *pHAE:Venus-H2B* (d), *pHSL2:Venus-H2B* and (**e**), *pFLS2:Venus-H2B* in 7 days-old roots. Representative pictures of n=8. Scale bar = 50 μm, single plane image, magenta = propidium iodide stain.

The online version of this article includes the following video and figure supplement(s) for figure 1:

**Figure supplement 1.** IDA induces ROS production in *Arabidopsis*.

**Figure supplement 2.** mIDA-induced $[Ca^{2+}]_{cyt}$ release in *Arabidopsis* roots.

**Figure supplement 3.** flg22-induced $[Ca^{2+}]_{cyt}$ release in *Arabidopsis* roots.

**Figure supplement 4.** The IDA triggered increase in $[Ca2+]_{cyt}$ is abolished in the presence of $Ca^{2+}$ inhibitors.

**Figure 1—video 1.** $[Ca^{2+}]_{cyt}$ dynamics in R-GECO1 expressing root tip in response to 1 μM mIDA.
https://elifesciences.org/articles/87912/figures#fig1video1

**Figure 1—video 2.** $[Ca^{2+}]_{cyt}$ dynamics in R-GECO1 expressing root tip in response to 1 μM flg22.
https://elifesciences.org/articles/87912/figures#fig1video2

**Figure 1—video 3.** $[Ca^{2+}]_{cyt}$ dynamics in R-GECO1 expressing root tip in response to 1 mM eATP.
https://elifesciences.org/articles/87912/figures#fig1video3

The $[Ca^{2+}]_{cyt}$ response in roots to the bacterial elicitor flg22 has previously been studied using the R-GECO1 sensor (*Keinath et al., 2015*). To better understand the specificity of the mIDA induced $Ca^{2+}$ response, we aimed to compare the $Ca^{2+}$ dynamics in mIDA-treated roots to those treated with flg22. We observed striking differences in the onset and distribution of the $Ca^{2+}$ signals. Analysis of roots treated with 1 μM flg22 showed that the $Ca^{2+}$ signal initiated in the root elongation zone from where it spread toward the meristematic zone as a single wave and that the signal amplitude was at a

maximum within the elongation zone and decreased as the signal spread (*Figure 1b*, *Figure 1—video 2*). These observations indicate differences in tissue specificity of Ca²⁺ responses between mIDA and flg22, which we hypothesize depend on the cellular distribution of their cognate receptors. Indeed, when investigating the promoter activity of the *HAE*, *HSL2*, and *FLS2* receptors by the use of nuclear localized transcriptional reporter lines we observed a different pattern of fluorescent nuclei in the roots (*Figure 1c, d and e*). *pHAE:Venus-H2B* lines had fluorescent expression in the epidermis and stele of the elongation zone (*Figure 1c*) and *pHSL2:Venus-H2B* lines in the lateral root cap, root tip and root meristem (*Figure 1d*); while fluorescent nuclei were observed in the stele of the elongation zone in *pFLS2:Venus-H2B lines* (*Figure 1e*). The different patterns of fluorescent nuclei observed for the three different receptor constructs could indeed explain the differences in the Ca²⁺ signatures triggered by mIDA and flg22. However, the receptors promoter activity show a broader expression pattern compared to the respective ligand induced [Ca²⁺]_cyt responses, indicating additional regulating factors for the observed [Ca²⁺]_cyt response.

To further investigate the mIDA induced Ca²⁺ response, we used a cytosolic localized Aequorin-based luminescence Ca²⁺ sensor (Aeq) (*Knight et al., 1991*). The Aeq sensor was chosen due to the well-established use of this sensor in studies investigating peptide induced Ca²⁺ responses (*Ranf et al., 2011*) Removing the last Asparagine of the mIDA peptide (IDAΔN69; *Supplementary file 1*) renders it inactive (*Butenko et al., 2014*). Seven days old *Aeq*-Seedlings treated with 1 µM IDAΔN69 did not show any increase in [Ca²⁺]_cyt (*Figure 1—figure supplement 4a*). To investigate if the [Ca²⁺]_cyt increase triggered by mIDA was dependent on extracellular Ca²⁺, we pre-treated seedling with LaCl₃ and Ethylene glycol-bis (2-aminoethylether)- N, N, N′,N′-tetraacetic acid (EGTA). LaCl₃ and EGTA are inhibitors that block PM localized cation channels and chelate Ca²⁺ in the extracellular space, respectively (*Knight et al., 1997*). The mIDA induced response was abolished in *Aeq*-seedlings pre-incubated in

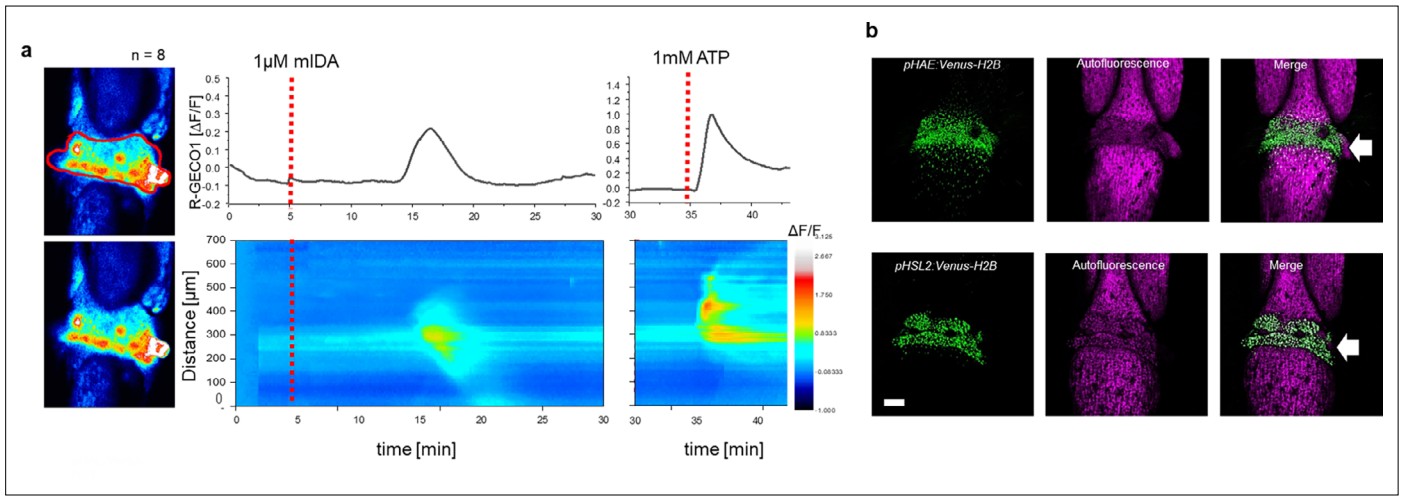

**Figure 2.** mIDA-induced [Ca²⁺]_cyt release in *Arabidopsis* abscission zones. (**a**) Normalized R-GECO1 fluorescence intensities (ΔF/F) were measured from regions of interest (ROI) (outlined in red) in floral abscission zone (AZ)s. Shown are cytosolic calcium concentration ([Ca²⁺]_cyt) dynamics in the ROI in response to 1 µM mIDA over time (see also *Figure 2—video 1*). Representative response from 8 flowers (*Figure 2—figure supplement 2*). Red lines at 5 min (min) indicates application of mIDA peptide or application of eATP at 35 min (for AZs), respectively. The increase in [Ca²⁺]_cyt response propagates through the AZ as a single wave. (**b**) Expression of *pHAE:Venus-H2B* and *pHSL2:Venus-H2B* in flowers at position 6 (See *Figure 2—figure supplement 1a* for positions; arrowhead indicates AZ). Representative pictures of n=8, scale bar = 100 µm, maximum intensity projections of z-stacks.

The online version of this article includes the following video and figure supplement(s) for figure 2:

**Figure supplement 1.** mIDA induces a Ca²⁺ response in flowers.

**Figure supplement 2.** mIDA-induced [Ca²⁺]_cyt release in *Arabidopsis* abscission zones.

**Figure supplement 3.** mIDAΔN69 does not induce a [Ca²⁺]_cyt release in *Arabidopsis* abscission zones (AZ).

**Figure supplement 4.** Investigation of CNGCs and RBOHs in the abscission process.

**Figure 2—video 1.** [Ca²⁺]_cyt dynamics in R-GECO1 expressing AZ in response to 1 µM mIDA.

https://elifesciences.org/articles/87912/figures#fig2video1

**Figure 2—video 2.** [Ca²⁺]_cyt dynamics in R-GECO1 expressing AZ in response to to 1 µM of the inactive IDA peptide, IDA^ΔN69.

https://elifesciences.org/articles/87912/figures#fig2video2

2 mM EGTA or 1 mM LaCL$_3$ (*Figure 1—figure supplement 4b*), indicating that the mIDA dependent [Ca$^{2+}$]$_{cyt}$ response depends on Ca$^{2+}$ from the extracellular space.

Next, we set out to investigate whether mIDA would trigger an increase in [Ca$^{2+}$]$_{cyt}$ in floral AZ cells. We utilized the Aeq expressing line and monitored [Ca$^{2+}$]$_{cyt}$ changes in flowers at different developmental stages (see *Figure 2—figure supplement 1a* for developmental stages [floral positions]) to 1 µM mIDA. Interestingly, only flowers at the stage where there is an initial weakening of the AZ cell walls showed an increase in [Ca$^{2+}$]$_{cyt}$ (*Figure 2—figure supplement 1*). Flowers treated with mIDA prior to cell wall loosening showed no increase in luminescence (*Figure 2—figure supplement 1*), indicating that the mIDA-triggered Ca$^{2+}$ release in flowers correlates with the onset of the abscission process and the increase in *HAE* and *HSL2* expression at the AZ (*Cai and Lashbrook, 2008*; *Patharkar and Walker, 2015*). Using plants expressing the R-GECO1 sensor we performed a detailed investigation of the mIDA induced [Ca$^{2+}$]$_{cyt}$ response in flowers at position 6 which is the position where initial cell wall loosening occurs. The AZ region was analyzed for signal intensity values and revealed that the Ca$^{2+}$ signal was composed of one wave (*Figure 2*, *Figure 2—video 1*). The signal initiated close to the nectaries and spread throughout the AZ and further into the floral receptacle. Flowers treated with 1 µM IDAΔN69 did not show any increase in [Ca$^{2+}$]$_{cyt}$ in the AZ or receptacle, while a clear Ca$^{2+}$ wave could be detected in the whole AZ, receptacle and proximal pedicel after treatment with eATP (*Figure 2—figure supplement 3*, *Figure 2—video 2*). Detailed investigation of *HAE* and *HSL2* promoter activity in flowers at position 6 shows restricted promoter activity to the AZ cells (*Figure 2b*). The observed promoter activity correlates with the observed mIDA-induced [Ca$^{2+}$]$_{cyt}$ response.

## The ROS producing enzymes, RBOHD and RBOHF, are not involved in the developmental process of abscission

ROS and Ca$^{2+}$ are secondary messengers that can play a role in developmental processes involving cell wall remodeling, but are also important players in plant immunity (*Kärkönen and Kuchitsu, 2015*). We therefore set out to investigate if the IDA induced production of ROS and increase in [Ca$^{2+}$]$_{cyt}$ function in the developmental process of abscission or if these signaling molecules form part of IDA modulated plant immunity.

The IDA induced Ca$^{2+}$ response depends on Ca$^{2+}$ from extracellular space (*Figure 1—figure supplement 4b*). Ca$^{2+}$ transport over the PM is enabled by a variety of Ca$^{2+}$ permeable channels, including the cyclic nucleotide gated channels (CNGCs). Various CNGCs are known to be involved in peptide ligand signaling in plants, including the involvement of CNGC6 and CNGC9 in CLAVATA3/embryo surrounding region 40 signaling in roots (*Breiden et al., 2021*), and CNGC17 In phytosulfokine signaling (*Ladwig et al., 2015*). Using publicly available expression data we identified multiple CNGCs expressed in the AZ of *Arabidopsis* (*Cai and Lashbrook, 2008*; *Figure 2—figure supplement 4a*, *Supplementary file 3*) and investigated if plants carrying mutations in the *CNGSs* genes expressed in AZs showed a defect in floral organ abscission. We observed no deficiency in floral abscission in the *cngc* mutants (*Figure 2—figure supplement 4b*), in contrast to what is observed in the *hae hsl2* mutant where all floral organs are retained throughout the inflorescence. Ca$^{2+}$ partially activates members of the NOX family of nicotinamide adenine dinucleotide phosphate (NADPH) oxidases (RBOHs), key producers of ROS (*Kadota et al., 2015*). Two members of this family, *RBOHD* and *RBOHF* have high transcriptional levels in the floral AZs (*Cai and Lashbrook, 2008*; *Figure 2—figure supplement 4a*, *Supplementary file 3*) and have been reported to be important in the cell separation event during floral abscission in *Arabidopsis* (*Lee et al., 2018*). We therefore set out to investigate if ROS production in AZ cells was dependent on RBOHD and RBOHF and to quantify to which degree these NADPH oxidases were necessary for organ separation. We treated AZs with the ROS indicator, H2DCF-DA, which upon contact with ROS has fluorescent properties, and observed ROS in both WT and to a lower extent in *rbohd rbohdf* flowers. In addition, we observed normal floral abscission in the *rbohd rbohf* double mutant flowers, indicating that the developmental progression of cell separation is not dependent on RBOHD and RBOHF (*Figure 2—figure supplement 4c and d*). These results are in stark contrast to what has previously been reported, where cell separation during floral abscission was shown to be dependent on RBOHD and RBOHF (*Lee et al., 2018*). We used a stress transducer to quantify the force needed to remove petals, the petal breakstrength (pBS; *Stenvik et al., 2008*), from the receptacle of gradually older flowers along the inflorescence of WT and *rbohd rbohf* flowers. Measurements showed a significant lower pBS value for the *rbohd*

*rbohf* mutant compared to WT at the developmental stage where cell loosening normally occurs, indicating premature cell wall loosening (*Figure 2—figure supplement 4e*). Furthermore, *rbohd rbohf* petals abscised one position earlier than WT (*Figure 2—figure supplement 4e*). Similar to *Crick et al., 2022*, we found no RBOHD and RBOHF dependent delay or absence of cell separation during floral abscission (*Crick et al., 2022*). Due to a known function of IDA in inducing ROS production (*Figure 1—figure supplement 1*), these results point towards a role for IDA in modulating additional responses to the process of cell separation.

### *IDA* is upregulated by biotic and abiotic factors

In *Arabidopsis*, infection with the pathogen *P. syringae* induces cauline leaf abscission. Interestingly, stress induced cauline leaf abscission was reduced in plants with mutations in components of the IDA-HAE/HSL2 signaling pathway (*Patharkar et al., 2017*; *Patharkar and Walker, 2016*). In addition, upregulation of defense genes in floral AZ cells during the abscission process is altered in the *hae hsl2* mutant (*Cai and Lashbrook, 2008*; *Niederhuth et al., 2013*), indicating that the IDA-HAE/HSL2 signaling system may be involved in regulating defense responses in cells undergoing cell separation. We further investigated if *IDA* is upregulated by biotic and abiotic elicitors. Transgenic plants containing the *β-glucuronidase (GUS)* gene under the control of the *IDA* promoter (*pIDA:GUS*) have previously been reported to exhibit *GUS* expression in cells overlaying newly formed LR primordia

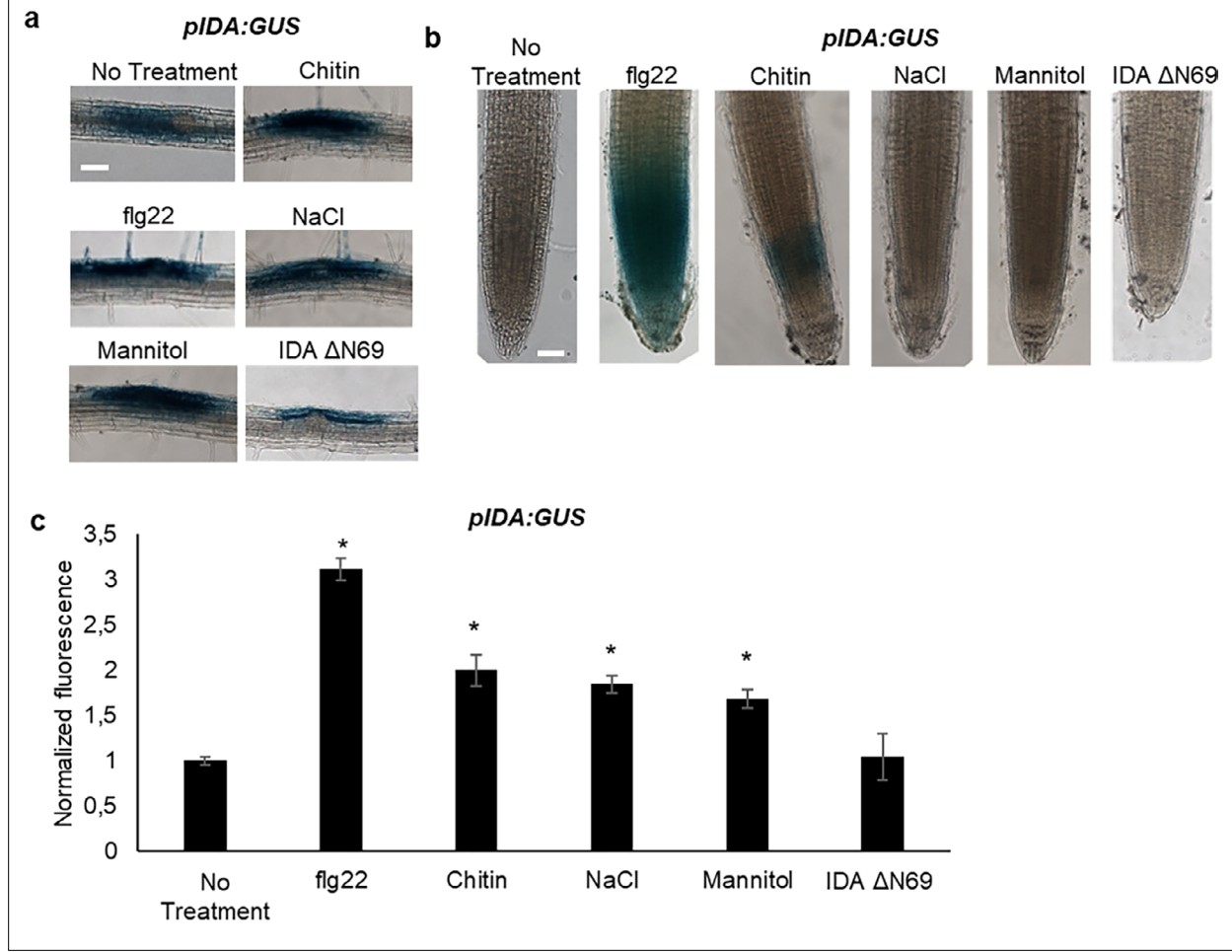

**Figure 3.** *IDA* is induced by biotic and abiotic stress. Representative pictures of *pIDA:GUS* expression after 12 hr treatment with 1 µM flg22, chitin, NaCl, Mannitol and 1 µM IDAΔN69 in (**a**) cells surrounding emerging lateral roots, and (**b**) in the main root. (**c**) Normalized emitted fluorescence of fluorochrome 4-methyl umbelliferone (4-MU) in 7 days-old seedlings after 12 hr treatment with 1 µM flg22, chitin, NaCl, Mannitol and 1 µM of the inactive IDA peptide, IDAΔN69. Normalized to 1 on No Treatment sample. Controls were not subjected to any stimuli (No treatment). n=6, experiment repeated three times. *=significantly different from the non-treated (No treatment) sample (p<0.05, student t-test, two tailed). Controls were not subjected to any stimuli (No treatment). (**a**, **b**) Representative picture of n=10, experiment repeated three times, scale bar = 50 µm.

(*Butenko et al., 2003*; *Kumpf et al., 2013*). To investigate the expression of the *IDA* gene in response to biotic stress, *pIDA:GUS* seedlings were exposed to the bacterial elicitor flg22 and fungal chitin. Compared to the control seedlings, the seedlings exposed to flg22 and chitin showed a significant increase in *GUS* expression in LR primordia (*Figure 3a*). Also, when comparing untreated root tips to those exposed to flg22 and chitin we observed a GUS signal in the primary root of flg22 and chitin treated roots (*Figure 3b*). We used the fluorogenic substrate 4-methylumbelliferyl β-D-glucuronide (4 MUG) which is hydrolysed to the fluorochrome 4-methyl umbelliferone (4-MU) to quantify GUS-activity (*Blázquez, 2007*). A significant increase in fluorescence was observed for flg22 and chitin treated samples compared to untreated controls, (*Figure 3b*), which was not observed when exposing seedlings to IDAΔN69 indicating that increased *IDA* expression is not triggered simply by the application of a peptide (*Figure 3*). Spatial *pIDA:GUS* expression was also monitored upon abiotic stress. When *pIDA:GUS* seedling were treated with the osmotic agent mannitol or exposed to salt stress by NaCl, both treatments simulating dry soil, a similar increment in GUS signal was observed surrounding the LR primordia (*Figure 3*). However, no *pIDA:GUS* expression was detected in the primary root (*Figure 3b*). Taken together these results indicate an upregulation of *IDA* in tissue involved in cell separation processes such as the LR primordia and, interestingly, the root cap upon biotic stress. The process of cell separation is a fast and time-restricted response where cells previously protected by outer cell layers are rapidly exposed to the environment. These cells need strong and rapid protection against the environment, and we therefore aimed to investigate if IDA is involved in modulating immune responses in addition to inducing ROS and $[Ca^{2+}]_{cyt}$ responses.

## mIDA can induce production of callose as a long-term defense response

We aimed to investigate a possible role of IDA in regulating long-term defense responses in the plant, such as the production of callose. Callose deposition is a resulting hallmark of the late plant immunity responses that together with other events such as stomatal closure and production of ethylene leads to inhibition of pathogen multiplication and containment of disease (reviewed in *Wang et al., 2021*). Callose deposition is highly increased in response to flg22 (*Gómez-Gómez et al., 1999*). Promoter activity studies of *pHAE* and *pHSL2* indicates a weak activity of the promoters in the cotelydons, (*Figure 1—figure supplement 1d*) and we thus treated WT seedlings with 1 µM mIDA and measured the callose depositions per area in the cotyledons. We detected no difference to water treated controls indicating that either callose deposition is not a cellular outcome for mIDA treatment (*Figure 4a and b*), or that receptor availability in the cotyledons is too low to observe a response We further tested if mIDA could induce callose depositions in cotyledons expressing the full length HAE receptor under a constitutive promoter (*35 S:HAE-YFP*). An increase in the callose depositions per area upon mIDA

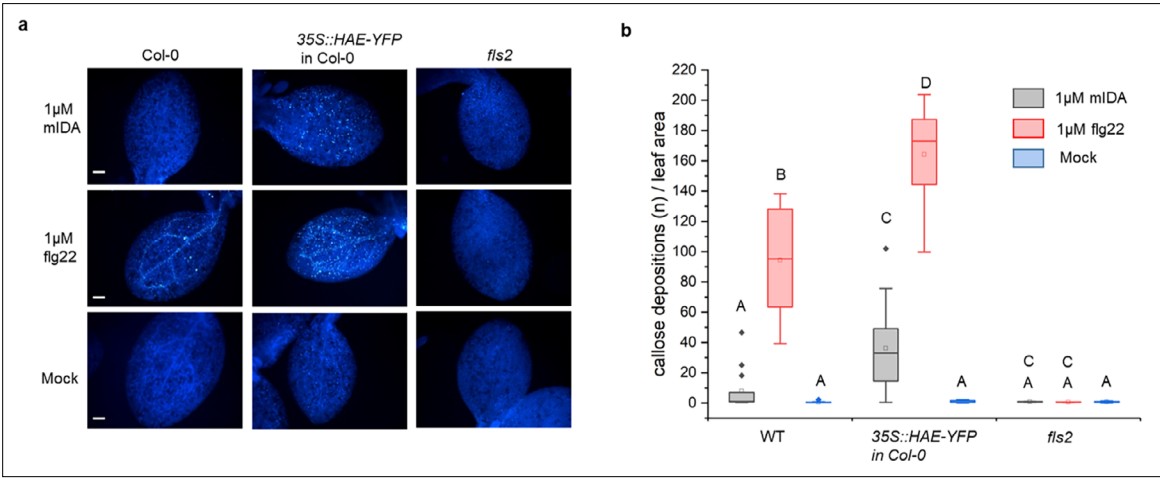

**Figure 4.** mIDA induce callose deposition in *Arabidopsis* cotelydons expressing 35 S::HAE-YFP. Callose deposition in Col-0 WT, 35 S:HAE in Col-0 and *fls2* treated with water (mock treatment), 1 µM mIDA or 1 uM flg22. (**a**) Callose deposition could be observed in cotyledons of eight day old Col-0 WT and *35 S:HAE in* seedlings treated with 1 µM flg22, and to a smaller extend in 35 S:HAE treated with 1 µM mIDA. No callose deposition could be detected in the *fls2* mutant. Representative images of 9–12 seedlings per genotype. Scale bar = 500 µm. (**b**) Total callose depositions for the different genotypes treated with water (mock treatment), 1 µM mIDA or 1 µM flg22. Statistically significant difference at (p<0.05). N=9–12.

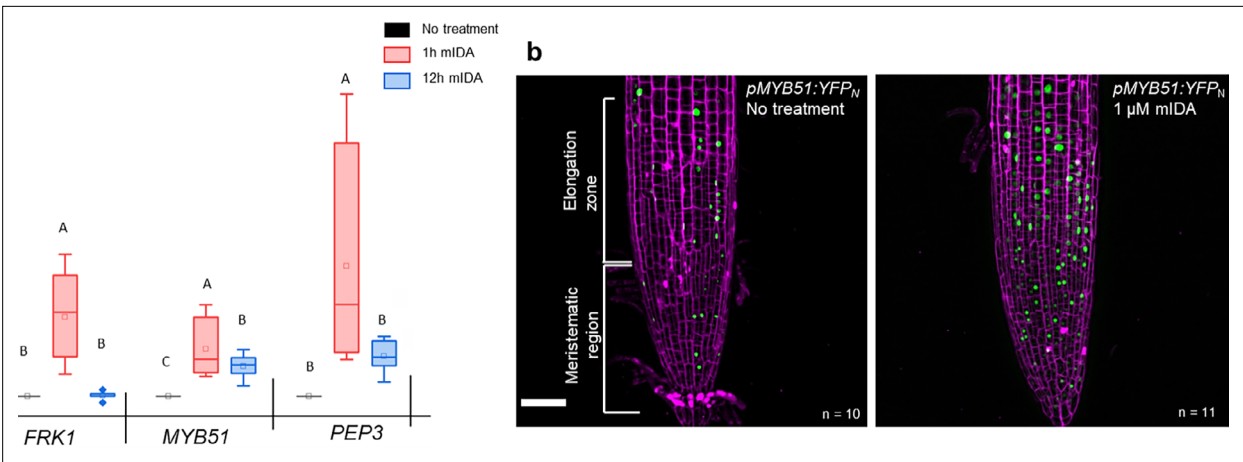

**Figure 5.** mIDA induced transcription of defense-associated marker genes. (**a**) Transcripts of *FRK1*, *MYB51*, and *PEP3* in WT Col-0 seedlings exposed to 1 µM mIDA for 1 hr (red) and to 1 µM mIDA for 12 hr (blue) compared to untreated tissue (black). RNA levels were measured by RT-qPCR analysis. *ACTIN* was used to normalize mRNA levels. The figure represents three biological replicates with four technical replicates. Statistical analyses was performed comparing induction times of individual genes using one-way ANOVA and post-hoc Tukey's test (p<0.05). (**b**) *pMYB51:YFP_N* expression is enhanced in roots after 7 hr exposure to 1 µM mIDA peptide compared to untreated roots (control), scale bar = 50 µm, maximum intensity projections of z-stacks, magenta = propidium iodide stain.

The online version of this article includes the following figure supplement(s) for figure 5:

**Figure supplement 1.** mIDA induced enhanced transcription of defense-associated marker genes is only partially dependent on the HAE and HSL2 receptors.

**Figure supplement 2.** mIDA and flg22 induced expression of *pMYB51:YFP*.

treatment compared to water treated controls was observed in these lines (*Figure 4a and b*). As a control, flg22-treated WT seedlings showed increased callose deposition, which was not observed in flg22-treated *fls2* seedlings (*Figure 4a and b*). These results indicate that mIDA, is capable of promoting a HAE-dependent deposition of callose as a long-term defense response.

## mIDA triggers expression of defense-associated marker genes associated with innate immunity

A well-known part of a plant immune response is the enhanced transcription of genes involved in immunity. Transcriptional reprogramming mostly mediated by WRKY transcription factors takes place during microbe-associated molecular pattern (MAMP)- triggered immunity and is essential to mount an appropriate host defense response (*Birkenbihl et al., 2017*). We selected well-established defense-associated marker genes: *FLG22-induced receptor-like kinase1* (*FRK1*), a specific and early immune-responsive gene activated by multiple MAMPs (*Asai et al., 2002*; *He et al., 2006*), *MYB51*, a WRKY target gene, previously shown to increase in response to the MAMP flg22 (*Frerigmann et al., 2016*) resulting in biosynthesis of the secondary metabolite indolic glucosinolates; and an endogenous danger peptide, ELICITOR PEPTIDE 3 (PEP3) (*Huffaker et al., 2006*). Seven days-old WT seedlings treated with 1 µM mIDA were monitored for changes in transcription of the aforementioned genes by RT-qPCR compared to untreated controls at two time points, 1 hr and 12 hr after treatment. All genes showed a significant transcriptional elevation 1 hr after mIDA treatment (*Figure 5a*). This in accordance with previous reports, where the expression of *FRK1* and *PEP3* was monitored in response to bacterial elicitors (*He et al., 2006*; *Huffaker et al., 2006*). After 12 h, *FRK1* and *PEP3* transcription in mIDA-treated tissue was not significantly different from untreated tissue, whereas transcription of *MYB51* in mIDA-treated tissue remained elevated after 12 hr but was significantly lower compared to 1 hr after mIDA treatment (*Figure 5a*). Interestingly, the observed increase in transcription of the defense genes *MYB51* and *PEP3*, was only partially reduced in the *hae hsl2* mutant (*Figure 5—figure supplement 1*). In contrast, *FRK1* showed higher transcriptional levels in the *hae hsl2* mutant compared to WT (*Figure 5—figure supplement 1*). These results indicate that also other receptors than HAE and HSL2 are involved in the mIDA-induced enhancement of the defense genes *FRK1*, *MYB51*, and *PEP3*.

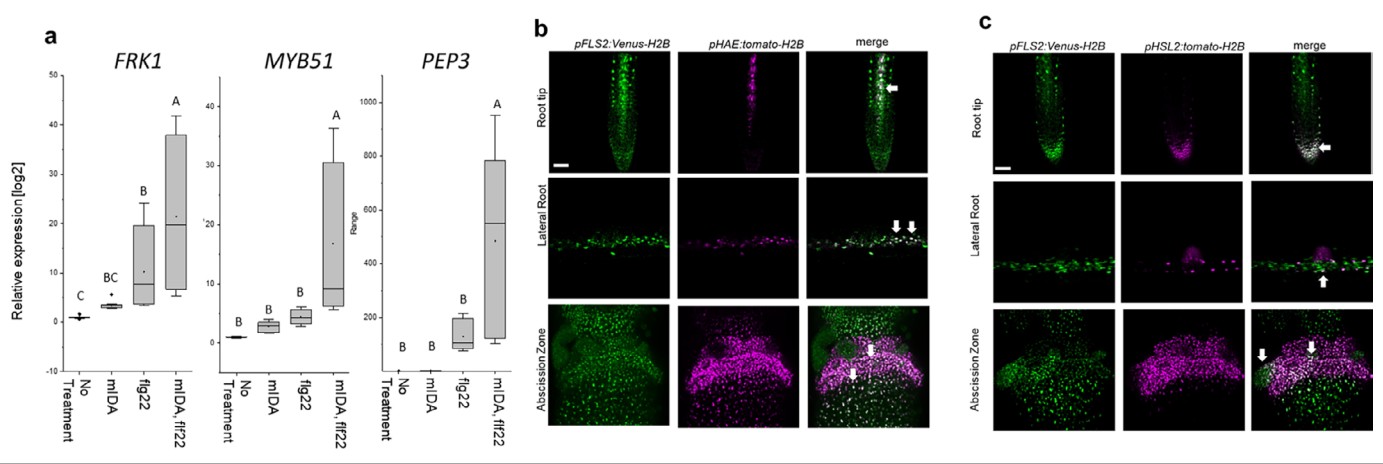

**Figure 6.** mIDA and flg22 co-treatment enhances the transcription of defense-associated marker genes. (**a**) RT-qPCR data showing transcription of *FRK1*, *MYB51*, and *PEP3* in Col-0 WT seedlings exposed to 1 μM mIDA, 1 μM flg22 or a combination of 1 μM mIDA and 1 μM flg22 for 1 hr compared to untreated seedlings (No treatment control). *ACTIN* was used to normalize mRNA levels. Figure represents three biological replicates with four technical replicates. Statistical analyses comparing No Treatment to peptide-treated samples was performed using one-way ANOVA and post-hoc Tukey's test (p<0.05). (**b**) Microscopic analysis of 7-day-old *pFLS2:Venus-H2B pHAE:Tomato-H2B* expressing plants of root tip (upper panel) and lateral root (middle panel) of 7 days old plants, and flowers at position 6 (lower panel). Fluorescent nuclei representing co-expression of the Venus and Tomato marker could be observed in cells surrounding emerging LRs, in the stele of the root, as well as in the abscission zone. (**c**) Microscopic analysis of 7-days-old *pFLS2:Venus-H2B pHSL2:Tomato-H2B* expressing plants of root tip (upper panel) and lateral root (middle panel) of 7 days old plants, and flowers at position 6 (lower panel). Fluorescent nuclei representing co-expression of the Venus and Tomato marker could be observed in the root tip, in cells surrounding emerging LRs, as well as in the abscission zone. Root pictures scale bar = 50 μm, single plane image. Abscission zone images = maximum intensity projections of z-stacks. Scale bar = 50 μm.

The online version of this article includes the following figure supplement(s) for figure 6:

**Figure supplement 1.** mIDA and PIP1 co-treatment do not enhances transcription of defense-associated marker genes.

We then analyzed a plant line expressing nuclear localized YFP from the MYB51 promoter (*pMYB51:YFP^N*) (*Poncini et al., 2017*) treated with mIDA. Enhanced expression of p*MYB51:^{YFPN}* was predominantly detected in the meristematic zone of the root after 7 hr of mIDA treatment, compared to a non-treated control (*Figure 5b*). Moreover, comparison of mIDA induction with the elicitor-triggered response of *pMYB51:YFP^N* to flg22 showed similar temporal expression in the root (*Figure 5—figure supplement 2*). Together, these data show that IDA can trigger a rapid increase in the expression of key genes involved in immunity. We next investigated if the mIDA induced transcription of defense related genes was similar to that induced by flg22. Seven days-old Col-0 WT seedlings were treated for 1 hr with mIDA, flg22 or a combination of mIDA and flg22 to explore potential additive effects. The relative increase in transcription of the genes tested is similar when seedlings are treated with mIDA and flg22 (*Figure 6a*). However, when treating seedlings with both peptides the relative transcription of the genes of interest increased substantially. To our surprise, the combined enhanced transcription when seedling where co-treated with both flg22 and mIDA exceeds a simple additive effect of the two peptides. To investigate if the observed increase is specific to a combination of mIDA and flg22, we investigated if co-treatment with mIDA and PIP1, an endogenous DAMP peptide known to amplify the immune response triggered by flg22 (*Hou et al., 2014*) gave a similar effect. Seedlings were treated for 1 hr with mIDA, PIP1 or a combination of the peptides to explore additive effects. In contrast to what we observed with flg22, no enhanced transcription of *FRK1*, *MYB51* and *PEP3* were observed after co-treatment with PIP1 and mIDA (*Figure 6—figure supplement 1*). These results suggest a role of IDA in enhancing the defense response triggered by flg22. To explore whether cells capable of undergoing cell separation in response to IDA and IDL peptides could be expressing *HAE* or *HSL2* in combination with *FLS2* we made transcriptional reporter lines expressing each of the receptors in fusion with either the nuclear localized YFP-derived fluorophore Venus protein or the RFP-derived Tomato protein (*pRECEPTOR:Venus/Tomato-H2B*) and crossed the lines to each other. Plants expressing both constructs were inspected for fluorescent nuclei in 7 day-old roots as well as in floral organs of floral position 7. Fluorescent nuclei with overlapping Venus and Tomato expression

were observed both in the vascular tissue of the root, in cells surrounding LRs as well as in cells of the AZs of plants expressing *pFLS2-Venus-H2B* and *pHAE-Tomat-H2B* (*Figure 6b*). Similar observations were done in the AZ of lines expressing *pFLS2-Venus-H2B* and *pHSL2-Tomat-H2B lines*; however, in these lines overlapping fluorescent nuclei in roots were mainly observed in the epidermal cells surrounding the LR and in the root tip, as well as in the root cap (*Figure 6c*). Based on these results, we propose a role for IDA in enhancing immune responses in tissue undergoing cell separation, possibly by enhancing cellular responses activated by immune receptors, such as FLS2.

## Discussion

Several plant species can abscise infected organs to limit colonization of pathogenic microorganisms thereby adding an additional layer of defense to the innate immune system (*Kissoudis et al., 2016*; *Patharkar et al., 2017*). Also, in a developmental context, there is an induction of defense-associated genes in AZ cells during cell separation and prior to the formation of the suberized layer that functions as a barrier rendering protection to pathogen attacks (*Cai and Lashbrook, 2008*; *Niederhuth et al., 2013*; *Roberts et al., 2000*; *Taylor and Walker, 2018*). Interestingly, the induction of defense genes during floral organ abscission is altered in *hae hsl2* plants (*Niederhuth et al., 2013*). It is largely unknown how molecular components regulating cell separation and how the IDA-HAE/HSL2 signaling pathway, contributes to modulation of plant immune responses. Here, we show that the mature IDA peptide, mIDA, is involved in activating known defense responses in *Arabidopsis*, including the production of ROS, the intracellular release of $Ca^{2+}$, the production of callose and the transcription of genes known to be involved in defense (*Figure 1—figure supplement 1*, *Figure 1*, *Figure 2*, *Figure 5*). We suggest a role for IDA in ensuring optimal cellular responses in tissue undergoing cell separation, regulating both development and defense.

The intracellular event occurring downstream of IDA is still partially unknown. Here, we report the release of cytosolic $Ca^{2+}$ as a new signaling component. Roots and AZs expressing the $Ca^{2+}$ sensor R-GECO1, showed a $[Ca^{2+}]_{cyt}$ in response to mIDA. Moreover, the increased $[Ca^{2+}]_{cyt}$ in response to mIDA in flowers occurred only at stages where cell separation was taking place, linking the observed $[Ca^{2+}]_{cyt}$ response to the temporal point of abscission and the need for an induced local defense response. During floral development the expression of the *HAE* and *HSL2* receptors increase prior to the onset of abscission (*Cai and Lashbrook, 2008*) and may account for the temporal mIDA induced $[Ca^{2+}]_{cyt}$ response observed.

Based on these results, it is likely that the expression of the receptors is the limiting factor for a cells ability to respond to the mIDA peptide with a $[Ca^{2+}]_{cyt}$ response. However, promoter activity of the receptors are observed in a broader area of the root than the observed $[Ca^{2+}]_{cyt}$ response. As an example, *pHSL2* is found to have a localized activation in the root tip, and when HSL2 is activated by IDL1 be involved in the regulation of cell separation during root cap sloughing (*Shi et al., 2018*). IDL1 and IDA have a high degree of similarity in the protein sequence, and *IDL1* expressed under the *IDA* promoter can fully rescue the abscission phenotype observed in the *ida* mutant (*Stenvik et al., 2008*). We can assume that IDL1 activates similar signaling components in the root cap as those used by IDA. Despite the expression of HSL2 and the involvement of the receptor in root cap sloughing, no $[Ca^{2+}]_{cyt}$ response is observed in this region when plants are treated with the IDA peptide. The absence of an observed $[Ca^{2+}]_{cyt}$ response in the root tip indicates that $[Ca^{2+}]_{cyt}$ is most likely not involved in the cell separation process during root cap sloughing, or that cells involved in this process only are able to induce a $[Ca^{2+}]_{cyt}$ response at a specific developmental time point. Thus, in addition to receptor availability, other factors must be involved in regulating the $[Ca^{2+}]_{cyt}$ response. It is intriguing that the mIDA cellular output is similar to that of flg22 (*Kadota et al., 2015*; *Kadota et al., 2014*) yet shows some distinct differences. The spatial distribution of the $Ca^{2+}$ wave originated by mIDA differs from that of flg22. The *rbohd rbohf* mutant shows ROS production in the AZ, indicating that other NADPH oxidases or cell wall peroxidases expressed in the AZ may be involved in ROS production, in contrast to the importance of RBOHD RBOHF in flg22 signaling (*Kadota et al., 2015*). This suggests that while IDA and flg22 share many components for their signal transduction, such as the BAK1 co-receptor (*Chinchilla et al., 2007*; *Meng et al., 2016*) and MAPK cascade (*Asai et al., 2002*; *Cho et al., 2008*), there are other specific components that are likely important to provide differences in signaling. Despite the observation of the numerous defense related responses to mIDA, we cannot exclude that the mIDA induced $[Ca^{2+}]_{cyt}$ and rises in ROS levels may also have an intrinsic developmental role such

as modulating cell wall properties and cell expansion similar to what is observed during FERONIA (FER) signaling (*Dünser et al., 2019*; *Feng et al., 2018*). To further decipher the importance of the mIDA-induced $Ca^{2+}$ and ROS in defense, it will be important to identify the specific signaling components responsible for this cellular output in addition to investigating the susceptibility of mutants to pathogen exposure.

Interestingly, FLS2 arrangement into nanoscale domains at the PM is dependent on FER. FLS2 becomes more disperse and mobile in *fer* mutants, implying a role for FER in regulating FLS2 activity (*Gronnier et al., 2022*). Also altering of the cell wall affects FLS2 nanoscale organization (*McKenna et al., 2019*). Formation of nanoscale domains containing the HAE and HSL2 receptors may be an important signaling step in the IDA-HAE/HSL2 signaling pathway. From our promoter analysis, we clearly see overlapping activity of the *HAE* and *HSL2* promoters with the *FLS2* promoter in planta, providing the possibility that the receptors are localized at the PM of the same cells and can have coordinated signaling events. An intriguing thought is a possible co-localization of HAE and HSL2 with FLS2 in nanoscale domains, making a signaling unit possible to involve multiple different receptors responsible for the signaling outcome observed. In this paper, we show that co-treatment with mIDA and flg22 induces an enhanced transcription of defense-related genes. How FLS2, HAE and HSL2 communicate at the PM of the responsive cells may give the answers to the molecular mechanisms behind the augmented transcriptional regulation observed when co-treatment of mIDA and flg22 is performed (*Figure 6*). We can investigate possible interaction partners of HAE and HSL2 looking at available data from a sensitized high-throughput interaction assay between extracellular domains (ECDs) of 200 Leucine-Rich Repeat (LRR) (*Smakowska-Luzan et al., 2018*). Interestingly, LRR-RLKs known to play a function in biotic or abiotic stress responses, such as receptor-like kinase 7 (RLK7), strubbelig-receptor family 3, receptor-like protein kinase 1 and FRK1, are found to interact mainly with HSL2, whereas HAE shows interaction with LRRs mainly involved in development (*Figure 7—figure supplement 1*). RLK7 has recently been identified as an additional receptor for CEP4, involved in regulating cell surface immunity and response to nitrogen starvation, indication a role of RLK7 as an additional receptor regulating defense responses in known peptide-receptor complexes (*Rzemieniewski et al., 2022*). It would be interesting to investigate a possible role of RLK7 in IDA-HAE/HSL2 signaling as a signaling partner in the observed IDA-induced defense responses.

In this paper, we propose a model where the IDA-HAE/HSL2 signaling pathway modulates defense responses in tissues undergoing cell separation. It is beneficial for the plant to ensure an upregulation of defense responses in tissues undergoing cell separation as a precaution against pathogen attack. This tissue is a major entry route for pathogens, and by combining the need of both the IDA peptide and the pathogen molecular pattern, such as flg22, for the induction of a strong defense response, the plant ensures maximum immunity in the most prone cells without spending energy inducing this in all cells (*Figure 7*). Such a molecular mechanism, where the presence of mIDA in tissue exposed to stress leads to maximum activation of the immune responses, will ensure maximal protection of infected cells during cell separation, cells which are major potential entry routes for invading pathogens.

## Methods

### Accession numbers of genes studied in this work

HSL2 At5g65710, IDA At1g68765, FLS2 At5g46330, HAE At4g28490, MYB51 At1g18570, RBOHD At5g47910, RBOHF At1g64060, PEP3 At5g64905, WRKY33 At2g38470, FRK1 At2g19190. Plant lines used in this work: Ecotype Colombia-0 (Col-0) was used as wild type (WT). Mutant line: *hae* (SALK_021905), *hsl2*, (SALK_030520), *fls2* (SALK_062054), *rbohd* (SALK_070610), *rbohf* (SALK_059888), *cngc1* (SAIL_443), *cngc2* (SALK_066908), *cngc4* (SALK_081369), *cngc5* (SALK_149893), *cngc6* (SALK_064702), *cngc9* (SAIL_736), *cngc12* (SALK_093622). SALK lines were provided from Nottingham *Arabidopsis* Stock Centre (NASC).

### Plant lines

The *pIDA:GUS*, *pHSL2:Venus-H2B* and *pMYB51:YFP*$_N$ lines have been described previously (*Kumpf et al., 2013*; *Poncini et al., 2017*; *Shi et al., 2018*).The promoters of *HAE* (1601 bp *Kumpf et al., 2013*) and *HSL2* (2300 bp *Stø et al., 2015*) were available in the pDONRZeo vector (Thermo Fischer Scientific). Sequences corresponding to the *FLS2* promoter (988 bp *Robatzek et al., 2006*) were

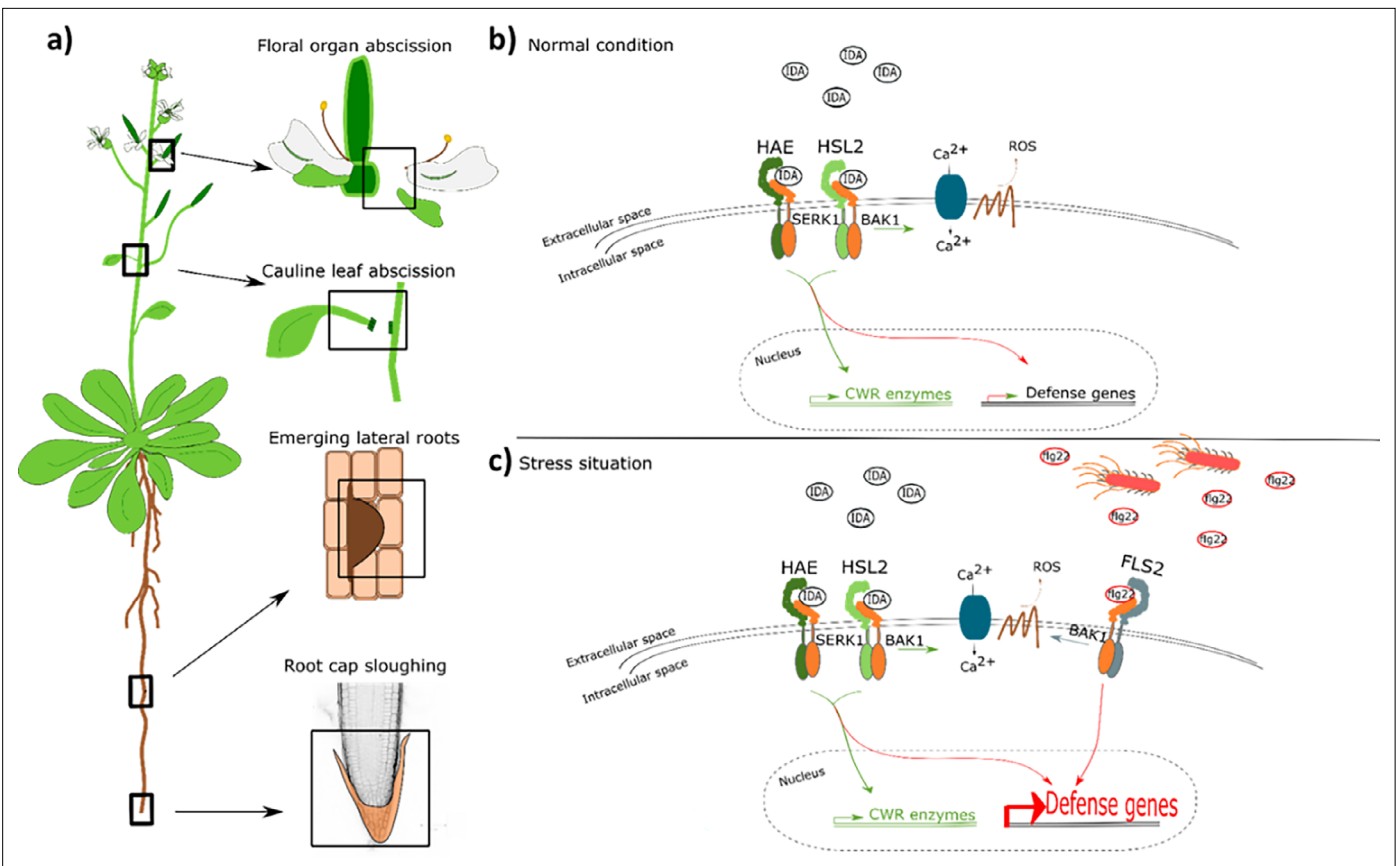

**Figure 7.** IDA regulates cell separation processes, but is also involved in major transcription of defense genes upon pathogen attack. (**a**) IDA and the IDL peptides control cell separation processes during plant development and in response to abiotic and biotic stress (*Butenko et al., 2003*; *Patharkar and Walker, 2016*; *Patharkar et al., 2017*; *Shi et al., 2018*). Tissue undergoing cell separation includes floral organ abscission, cauline leaf abscission, emerging of lateral roots and root cap sloughing. (**b**) During normal conditions, IDA control floral organ abscission and emergence of lateral roots by relaying a signal through receptor complexes including HAE, HSL2, SERK1 and BAK1 to modulate the expression of cell wall remodeling (CWR) genes as well as moderately expression of defense genes. IDA activates a receptor dependent production of ROS and an increase in $[Ca^{2+}]_{cyt}$. (**b**) Upon stress, such as pathogen attack, the activation of HAE and HSL2 acts in addition to activation of defense related receptors, such as FLS2, to enhance the expression of defense related genes significantly. This ensure optimal protection of cells undergoing cell separation, which may be major entry routes during a pathogen attack.

The online version of this article includes the following figure supplement(s) for figure 7:

**Figure supplement 1.** HAE (yellow) and HSL2 (red) display very distinct repertoire of immune and growth-related interacting LRR-RKs.

amplified from WT DNA (primers are listed in *Supplementary file 2*) and cloned into the pDONRZeo vector (Thermo Fischer Scientific). All promoter constructs were further recombined into the promotor:Venus (YFP)-H2B and Tomato-H2B destination vectors (*Somssich et al., 2016*) using the Invitrogen Gateway cloning system (Thermo Fischer Scientific). Constructs were transformed into *Agrobacterium tumefaciens* (*A. tumefaciens*) C58 and the floral dip method (*Clough and Bent, 1998*) was used to generate transgenic lines. Single-copy homozygous plant lines were selected and used in this study. The CDS of *HAE* was cloned into the pEarleyGate101 destination vector (*Earley et al., 2006*) using the Invitrogen Gateway cloning system and transformed into *A. tumefaciens* C58 and further used to generate the *35 S:HAE:YFP* lines.

## Growth conditions

Plants were grown in long day conditions (8 hr dark and 16 hr light) at 22 °C. Seeds were surface sterilized and plated out on MS-2 plates, stratified for 24 hr at 4 °C and grown on plates for 7 days before transferred to soil.

### Peptide sequences

Peptides used in this study were ordered from BIOMATIK. Peptide sequences are listed in **Supplementary file 1**.

### Primers

Primers for genotyping and generation of constructs were generated using VectorNTI. Gene specific primers for RT-qPCR were generated using Roche Probe Library Primer Design. All primers are listed in **Supplementary file 2**.

### Histochemical GUS assay

Seven days-old seedlings were pre-incubated for 12 hr in liquid MS-2 medium containing stimuli of interest; 1 µM peptide (**Supplementary file 1**), 60 mM Mannitol (M4125 – Sigma), 20 µg/mL Chitin (C9752 – Sigma), 50 mM NaCl and then stained for GUS activity following the protocol previously described (**Stenvik et al., 2008**). Roots were pictured using a Zeiss Axioplan2 microscope with an AxioCam HRc, 20 x air objective. The assay was performed on 10 individual roots and the experiment was repeated three times.

### Fluorescent GUS assay

Seven days-old seedlings were pre-incubated for 12 hr in liquid MS-2 medium with or without stimuli of interest; 1 µM peptide (**Supplementary file 1**), 60 mM Mannitol (M4125 – Sigma), 20 µg/mL Chitin (C9752 – Sigma), 50 mM NaCl. After treatment, 10 seedlings were incubated in wells containing 1 mL reaction mix as described in **Blázquez, 2007** containing: 10 mM EDTA (pH 8.0), 0.1 % SDS, 50 mM Sodium Phosphate (pH 7.0), 0.1 % Triton X-100, 1 mM 4 MUG (, M9130-Sigma) and incubated at 37 °C for 6 hr. Six 100 µl aliquots from each well were transferred to individual wells in a microtiter plate and the reaction was stopped by adding 50 µl of stop reagent (1 M Sodium Carbonate) to each well. Fluorescence was detected by the use of a Wallac 1420 VICTOR2 microplate luminometer (PerkinElmer) using an excitation wavelength of 365 nm and a filter wavelength of 430 nm. Each experiment was repeated three times.

### Confocal laser microscopy of roots and flowers expressing promoter:Venus/YFP-H2B and promoter:Tomato-H2B construct

Imaging of 7-day-old roots was performed on a LSM 880 Airyscan confocal microscope equipped with two flanking PMTs and a central 32 array GaAsP detector. Images were acquired with Zeiss Plan-Apochromat 20 x/0.8 WD = 0.55 M27 objective and excited with laser light of 405 nm, 488 nm and 561 nm. Roots were stained by 1 µM propidium iodide for 10 min and washed in dH$_2$O before imaging. Imaging of flowers was performed on an Andor Dragonfly spinning disk confocal using an EMCCD iXon Ultra detector. Images were acquired with a 10 x plan Apo NA 0.45 dry objective and excited with laser light of 405 nm, 488 nm, and 561 nm. Maximum intensity projections of z-stacks were acquired with step size of 1.47 µm. Image processing was performed in FIJI 51. These steps are: background subtraction, gaussian blur/smooth, brightness/contrast. Imaging was performed at the NorMIC Imaging platform.

### Calcium imaging using the R-GECO1 sensor

$[Ca^{2+}]_{cyt}$ in roots were detected using WT plants expressing the cytosolic localized single-fluorophore based $Ca^{2+}$ sensor, R-GECO1 in Col-0 background (**Keinath et al., 2015**). Measurements were performed using a confocal laser scanning microscopy Leica TCS SP8 STED 3 X using a 20 x multi NA 0.75 objective. Images were recorded with a frame rate of 5 s at 400 Hz. Seedling mounting was performed as described in **Krebs and Schumacher, 2013**. The plant tissues were incubated overnight in half strength MS, 21 °C and continuous light conditions before the day of imaging. $[Ca^{2+}]_{cyt}$ in the abscission zone were detected using WT plants expressing R-GECO1 (**Keinath et al., 2015**). Abscission zones of position 7 were used. Mounting of the abscission zones were performed using the same device as for seedlings (**Krebs and Schumacher, 2013**). The abscission zones were incubated in half strength MS medium for 1 hr before imaging. Measurements were performed with a Zeiss LSM880 Airyscan using a Plan-Apochromat 10×air NA 0.30 objective. Images were recorded with a frame rate of 10 s. R-GECO1 was excited with a white light laser at 561 nm and its emission was detected at 590

nm to 670 nm using a HyD detector. Laser power and gain settings were chosen for each experiment to maintain comparable intensity values. For mIDA and flg22 two-fold concentrations were prepared in half strength MS. mIDA or flg22 were added in a 1:1 volume ratio to the imaging chamber (final concentration 1 µM). ATP was prepared in a 100-fold concentration in half strength MS and added as a last treatment in a 1:100 volume ratio (final concentration 1 mM) to the imaging chamber as a positive control for activity of the R-GECO1 sensor. Image processing was performed in FIJI. These steps are: background subtraction, gaussian blur, MultiStackReg v1.45 (http://bradbusse.net/sciencedownloads.html), 32-bit conversion, threshold. Royal was used as a look up table. Fluorescence intensities of indicated ROIs were obtained from the 32-bit images (*Krebs and Schumacher, 2013*). Normalization was done using the following formula $\Delta F/F = (F-F0)/F0$ where F0 represents the mean of at least 1 min measurement without any treatment. R-GECO1 measurements were performed at the Center for Advanced imaging (CAi) at HHU and at NorMIC Imaging platform at the University of Oslo.

## Calcium measurements using the Aequorin (pMAQ2) sensor

$[Ca^{2+}]_{cyt}$ in seedlings and flowers were detected using WT plants expressing *p35S-apoaequorin* (*pMAQ2*) located to cytosol (Aeq) (*Knight et al., 1991*; *Ranf et al., 2011*). Aequorin luminescence was measured as previously described (*Ranf et al., 2012*). Emitted light was detected by the use of a Wallac 1420 VICTOR2 microplate luminometer (PerkinElmer). Differences in Aeq expression levels due to seedling size and expression of sensor were corrected by using luminescence at specific time point (L)/Max Luminescence (Lmax). Lmax was measured after peptide treatment by individually adding 100 µL 2 M $CaCl_2$ to each well and measuring luminescence constantly for 180 s (*Ranf et al., 2012*). Two M $CaCl_2$ disrupts the cells and releases the Aeq sensor into the solution where it will react with $Ca^{2+}$ and release the total possible response in the sample (Lmax) in form of a luminescent peak. A final concentration of 1 µM mIDA was added to each wells at the start of measurements. For inhibitor treatments, Aeq-seedlings were incubated in in 2 mM EGTA (Sigma-Aldrich) or 1 mM $LaCL_3$ (Sigma-Aldrich) O/N before measurements. For seedlings, three independent experiments were performed with 12 replications in each experiment. For flowers, three independent experiments were performed with four to six replications in each experiment.

## Measurements of reactive oxygen species (ROS)

ROS production was monitored by the use of a luminol-dependent assay as previously described (*Butenko et al., 2014*) using a Wallac 1420 VICTOR2 microplate luminometer (PerkinElmer). *Arabidopsis* leaves expressing *35 S:HAE:YFP* were cut into leaf discs and incubated in water overnight before measurements. A final concentration of 1 µM mIDA was added to each well at the start of measurements. All measurements were performed on six leaf discs and each experiment was repeated three times.

## ROS stain (H2DCF-DA)

Flowers at position 6 were gently incubated in staining solution (25 µM (2',7'-dichlorodihydrofluorescein diacetate) (H2DCF-DA) (Sigma-Aldrich, D6883), 50 mM KCL, 10 mM MES) for 10 min and further washed three times in wash solution (50 mM KCL, 10 mM MES). Imaging was done using a Dragonfly Airy scan spinning disk confocal microscope, excited by a 488 nm laser. A total of nine flowers per genotype were imaged. The experiment was repeated two independent times.

## Callose deposition staining and quantification

Callose deposition experiments were conducted as per *Luna et al., 2011* with modifications. Seeds from WT and *fls2* were vapor-phase sterilized for 4 hr. Seeds were transferred to 12-well plates containing 1 mL of filter-sterilized MS medium and 0.5 % MES with a final pH of 5.7, and stratified in the dark at 4 °C for 2 days. Plates were consecutively transferred to a growth chamber with 16 hr of light (150µE m-2 s-1)/8 hr darkness cycle at 22 °C. On the seventh day, growth medium was refreshed under sterile conditions. On the eight day, the treatments were applied: addition of 10 µL of water (mock); 10 µL of 100 µM flg22; or 10 µL of 100 µM mIDA (final peptide concentrations of 1 µM). After 24 hr the growth medium was replaced with 95 % ethanol and incubated overnight. Ethanol was replaced with an 8 M NaOH softening solution and incubated for 2 hr. Seedlings were washed three times in water and immediately transferred to the aniline blue staining

solution (100 mM KPO4-KOH, pH 11; 0.01 % aniline blue). After 2 hr of staining, the seedlings were mounted on 50 % glycerol and imaged under a ZEISS epifluorescence microscope with UV filter (BP 365/12 nm; FT 395 nm; LP 397 nm). Image analysis was performed on FIJI (*Schindelin et al., 2012*). In brief, cotyledons were cropped and measured the leaf area. Images were thresholded to remove autofluorescence and area of depositions measured to calculate the ratio of callose deposition area per cotyledon area unit. Between 9 and 12 seedlings were analyzed per genotype x treatment combination.

## Real-time quantitative PCR (RT-qPCR)

Seven days-old *Arabidopsis* seedlings (WT, *hae hsl2*) grown vertically on ½ sucrose MS-2 plates were transferred to liquid ½ MS-2 medium (non-treated) and liquid ½ MS-2 medium containing 1 µM peptide (*Supplementary file 1*) and incubated in growth chambers for 1 hr or 12 hr. Seedlings were flash-frozen in liquid nitrogen before total RNA was extracted using SpectrumTM Plant Total RNA Kit (Sigma- Aldrich). cDNA synthesis was performed as previously described (*Grini et al., 2009*). RT-qPCR was performed according to protocols provided by the manufacturer using FastStart Essential DNA Green Master (Roche) and LightCycler96 (Roche) instrument. ACTIN2 was used to normalize mRNA levels as described in *Grini et al., 2009*. Two-three biological replicates and 4 technical replicates including standard curves were performed for each sample.

## Petal Break strength (pBS)

The force required to remove a petal at a given position on the inflorescence was measured in gram equivalents using a load transducer as previously described (*Stenvik et al., 2008*). Plants were grown until they had at least 15 positions on the inflorescence. A minimum of 15 petals per position were measured. pBS measurements were performed on WT and *rbohd rbohf* (SALK_070610 SALK_059888) plants.

## Statistical methods

Two tailed students t-test (-<0.05) was used to identify significant differences in the fluorescent GUS assay by comparing treated samples to untreated samples of the same plant line. Statistical analysis of the RT-qPCR results was performed on all replicas using one-way or two-way ANOVA (as stated in the figure text) and post-hoc Tukey's test (p<0.05). Two tailed students t-test with (p<0.05) was used in the petal break strength (pBS) measurements to identify significant differences from WT at a given position on the inflorescence.

## Acknowledgements

We thank MK Anker, IM Stø, V Iversen and R Falleth for technical assistance in the laboratory and phytotrone. We thank the NorMic Imaging platform for the use and technical support. This work was supported by the Research Council of Norway (grant 230849) to VO Lalun and MA Butenko. Work by R Simon and M Breiden was supported through CEPLAS.

## Additional information

### Funding

| Funder | Grant reference number | Author |
| --- | --- | --- |
| Research Council of Norway | 230849 | Vilde Olsson Lalun Melinka A Butenko |
| Cluster of Excellence on Plant Sciences | | Maike Breiden Rüdiger GW Simon |

The funders had no role in study design, data collection, and interpretation, or the decision to submit the work for publication.

## Author contributions

Vilde Olsson Lalun, Conceptualization, Data curation, Formal analysis, Investigation, Visualization, Methodology, Writing – original draft, Writing – review and editing, Generated Arabidopsis lines and constructs, tested IDA expression to biotic and abiotic stress, performed gene expression studies, callose deposition assays, phenotypic analysis of mutants and ROS measurements, performed Ca2+ measurements, designed experiments, analyzed data, and drafted the manuscript and wrote the paper with input from all authors; Maike Breiden, Data curation, Formal analysis, Methodology, Writing – review and editing, Performed Ca2+ measurements, designed experiments, analyzed data and gave input on the manuscript; Sergio Galindo-Trigo, Data curation, Methodology, Writing – review and editing, Designed experiments, analyzed data, and gave input on the manuscript; Elwira Smakowska-Luzan, Investigation, Visualization, Writing – review and editing; Rüdiger GW Simon, Conceptualization, Supervision, Funding acquisition, Project administration, Writing – review and editing, Provided funding, designed experiments, drafted the manuscript and gave input on the manuscript; Melinka A Butenko, Conceptualization, Resources, Supervision, Funding acquisition, Writing – original draft, Project administration, Writing – review and editing, Provided funding, designed experiments, analyzed data, and drafted the manuscript and wrote the paper with input from all authors

## Author ORCIDs

Vilde Olsson Lalun ⓘ https://orcid.org/0000-0002-0453-7062
Rüdiger GW Simon ⓘ https://orcid.org/0000-0002-1317-7716
Melinka A Butenko ⓘ https://orcid.org/0000-0003-0750-7018

Reviewer #1 (Public Review): https://doi.org/10.7554/eLife.87912.3.sa1
Reviewer #3 (Public Review): https://doi.org/10.7554/eLife.87912.3.sa2
Author response https://doi.org/10.7554/eLife.87912.3.sa3

# Additional files

## Supplementary files

• Supplementary file 1. Peptide sequences.

• Supplementary file 2. Primers sequences and function.

• Supplementary file 3. Relative expression of genes of the *CNGC* and *RBOH* gene families during the onset of abscission (data from *Cai and Lashbrook, 2008*). See *Figure 2—figure supplement 1a* for flower developmental stages (p2-p6). CNGC = CYCLIC NUCLEOTIDE GATED CHANNEL, RBOH = RESPIRATORY BURST OXIDASE PROTEIN, IDA = INFLORESCENCE DEFICIENT IN ABSCISSION, HSL2 = HAESA LIKE 2.

• MDAR checklist

## Data availability

Original microscopy and imaging data from confocal and spinning disk microscopy is accessible through BioStudies: https://www.ebi.ac.uk/biostudies/bioimages/studies/S-BIAD1102.

The following dataset was generated:

| Author(s) | Year | Dataset title | Dataset URL | Database and Identifier |
|---|---|---|---|---|
| Lalun VO, Breiden M, Galindo-Trigoo S, Smakowska-Luzan E, Simon R, Butenko MA | 2024 | A dual function of the IDA peptide in regulating cell separation and modulating plant immunity at the molecular level | https://www.ebi.ac.uk/biostudies/bioimages/studies/S-BIAD1102 | BioImage Archive, S-BIAD1102 |

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
