## [Editor Report · eLife assessment]

This manuscript presents **valuable** findings on the role of a plant peptide in coordinating developmental and immune responses signaling. The evidence supporting the claims, while mainly descriptive and somewhat limited due to the main conclusions being drawn from overexpression lines, is mostly **solid**. The findings are interesting, they align with existing models, and they are of relevance to plant pathologists and developmental biologists.

---

## [Referee Report · Reviewer #1 (Public Review)]

A descriptive manuscript investigating the ability of a peptide, implicated in development, to induce signalling responses indicative of immunity. The work clearly documents the ability of the synthetic peptide to induce these responses, and open future work to link this back to physiology.

Comments on revised version:

Congratulations to the authors for the improvements to the manuscript.

I still have reservations, as raised by other reviewers, about whether the outputs observed can definitively be classified as immune/defence outputs without assaying an impact upon microbial growth. Indeed, this is challenging to address as many of the outputs are shared by multiple pathways. This is especially the case here as the peptide could have different effects in different tissues or cells with different expression levels of the receptors (e.g. hypothetically - no expression = no effect; weak expression - cell wall loosening and susceptibility; high expression - strong response and 'defence' response). I do however appreciate that the authors have toned down some of the conclusions regarding the defence response and also they included further reference to outputs also being from developmental pathways.

---

## [Referee Report · Reviewer #3 (Public Review)]

Previously, it has been shown the essential role of IDA peptide and HAESA receptor families in driving various cell separation processes such as abscission of flowers as a natural developmental process, of leaves as a defense mechanism when plants are under pathogenic attack or at the lateral root emergence and root tip cell sloughing. In this work, Olsson et al. show for the first time the possible role of IDA peptide in triggering plant innate immunity after the cell separation process occurred. Such an event has been previously proposed to take place in order to seal open remaining tissue after cell separation to avoid creating an entry point for opportunistic pathogens. The elegant experiments in this work demonstrate that IDA peptide is triggering the defense-associated marker genes together with immune specific responses including release of ROS and intracellular CA2+. Thus, the work highlights an intriguing direct link between endogenous cell wall remodeling and plant immunity. Moreover, the upregulation of IDA in response to abiotic and especially biotic stimuli are providing a valuable indication for potential involvement of HAE/IDA signalling in other processes than plant development.

Comments on revised version:

We thank the authors for addressing our previous comments. Overall, we are satisfied with the improvements and appreciate the hard work that has gone into this manuscript. We wish you all the best on the further publication pathway.

---

## [Author Response]

The following is the authors’ response to the original reviews.

**eLife assessment**
This valuable study provides insights into the IDA peptide with dual functions in development and immunity. The approach used is solid and helps to define the role of IDA in a two-step process, cell separation followed by activation of innate defenses. The main limitation of the study is the lack of direct evidence linking signaling by IDA and its HAE receptors to immunity. As such the work remains descriptive but it will nevertheless be of interest to a wide range of plant cell biologists.

We thank the reviewers for thoroughly reading our manuscript. We have used their comments and suggestions- to improve the manuscript. Below is a response to the reviewer's comments.

**Public Reviews:**

**Reviewer #1 (Public Review):**
The paper titled 'A dual function of the IDA peptide in regulating cell separation and modulating plant immunity at the molecular level' by Olsson Lalun et al., 2023 aims to understand how IDAHAE/HSL2 signalling modulates immunity, a pathway that has previously been implicated in development. This is a timely question to address as conflicting reports exist within the field. IDL6/7 have previously been shown to negatively regulate immune signalling, disease resistance and stress responses in leaf tissue, however IDA has been shown to positively regulate immunity through the shedding of infected tissues. Moreover, recently the related receptor NUT/HSL3 has been shown to positively regulate immune signalling and disease resistance. This work has the potential to bring clarity to this field, however the manuscript requires some additional work to address these questions. This is especially the case as it contracts some previous work with IDL peptides which are perceived by the same receptor complexes.Can IDA induce pathogen resistance? Does the infiltration of IDA into leaf tissue enhance or reduce pathogen growth? Previously it has been shown that IDL6 makes plants more susceptible. Is this also true for IDA? Currently cytoplasmic calcium influx and apoplastic ROS as overinterpreted as immune responses - these can also be induced by many developmental cue e.g. CLE40 induced calcium transients. Whilst gene expression is more specific is also true that treatment with synthetic peptides, which are recognised by LRR-RKs, can induce immune gene expression, especially in the short term, even when that is not there in vivo function e.g. doi.org/10.15252/embj.2019103894.

We thank the reviewer for the concerns raised and agree that further experiments including pathogen assays would strengthen the link between IDA signaling and immunity and we plan for such experiments in future work. We have however, modified the discussion to include the possible role of IDA induced Ca2+ and ROS during development. We have recently published a preprint (accepted for publication in JXB) ( (Galindo-Trigo et al., 2023, https://doi.org/10.1101/2023.09.12.557497)) strengthening the link between IDA and defense by identifying WRKY transcription factors that regulate IDA expression through a Y1H assay.

This paper shows that receptors other than hae/hsl2 are genetically required to induce defense gene expression, it would have been interesting to see what phenotype would be associated with higher order mutants of closely related haesa/haesa-like receptors. Indeed recently HSL1 has been shown to function as a receptor for IDA/IDL peptides. Could the triple mutant suppress all response? Could the different receptors have distinct outputs? For example for FRK1 gene expression the hae hsl2 mutant has an enhanced response. Could defence gene expression be primarily mediated by HSL1 with subfunctionalisation within this clade?

We agree that it would be interesting to also include HSL1 in our studies. However, the focus of this study has been on HAE and HSL2 and we wanted to explore their role in IDA induced defense responses. Including HSL1 in these studies will require generation of multiple transgenic lines and repeating most of the experiments and are experiments we will consider in a follow up study together with pathogen assays (that would also address the main concern raised in the comment above). We have however, modified the text to include the known function of HSL1 and discuss the possibility of subfunctionalisation of this receptor clade.

One striking finding of the study is the strong additive interaction between IDA and flg22 treatment on gene expression. Do the authors also see this for co-treatment of different peptides with flg22, or is this unique function of IDA? Is this receptor dependent (HAE/HSL1/HSL2)?

This is a good question. Since our study focuses on the IDA signaling pathway we preferentially tested if the additive effect observed between flg22 and mIDA was also observed when mIDA was combined with another peptide involved in defense. The endogenous peptide PIP1, has previously been shown to amplify flg22 signaling (Hou et al 2014, doi:10.1371/journal.ppat.1004331 ). In this study it is shown that co-treatment with flg22 and PIP1 gives increased resistance to Pseudomonas PstDC3000 compared to when plants are treated with each peptide separately. In the same study, the authors also show reduced flg22 induce transcriptional activity of two defense related genes WRKY33 and PR in the receptor like kinase7 (rlk7) mutant (the receptor perceiving PIP1) (). To investigate whether PIP1 would give the same additive effect with mIDA as that observed between flg22 and mIDA, we co-treated seedlings with PIP1 and mIDA. We observed no enhanced transcriptional activity of FRK1, MYB51 and PEP3 in tissue from plants treated with both PIP1 and mIDA peptides compared to single exposure. These results are presented in supplementary figure 11. In conclusion we do not think mIDA acts as a general amplifier of all immune elicitors in plants.

It is interesting how tissue specific calcium responses are in response to IDA and flg22, suggesting the cellular distribution of their cognate receptors. However, one striking observation made by the authors as well, is that the expression of promoter seems to be broader than the calcium response. Indicating that additional factors are required for the observed calcium response. Could diffusion of the peptide be a contributing factor, or are only some cells competent to induce a calcium response?

It is interesting that the authors look for floral abscission phenotypes in cngc and rbohd/f mutants to conclude for genetic requirement of these in floral abscission. Do the authors have a hypothesis for why they failed to see a phenotype for the rbohd/f mutant as was published previously? Do you think there might be additional players redundantly mediating these processes?

It is a possibility that diffusion of the peptide plays a role in the observed response. In a biological context we would assume that the local production of the peptides plays an important role in the cellular responses. In our experimental setup, we add the peptide externally and we can therefore assume that the overlaying cells get in contact with the peptide before cells in the inner tissues and this could be affecting the response recorded However, our results show that there is a differences between flg22 and mIDA induced responses even when the application of the peptides is performed in the same manner, indicating that the difference in the response is not primarily due to the diffusion rate of the peptides but is likely due to different factors being present in different cells. To acquire a better picture of the distribution of receptor expression in the root tissue and to investigate in which cells the receptors have an overlapping expression pattern, we have included results in figure 6 showing plant lines co-expressing transcriptional reporters of FLS2 and HAE or HSL2.

Can you observe callose deposition in the cotyledons of the 35S::HAE line? Are the receptors expressed in native cotyledons? This is the only phenotype tested in the cotyledons.

We thank the reviewer for this valuable comment. We have now conducted callose deposition assay on the 35S:HAE line. And Indeed, we observe callose depositions when cotyledons from a 35S:HAE line is treated with mIDA. We have included these results in figure 4 and have adjusted the text regarding the callose assay accordingly. In addition, we have analyzed the promoter activity of pHAE in cotelydons and we observe weak promoter activity. These results are included as supplementary figure 1d.

Are flg22-induced calcium responses affected in hae hsl2?

The experiment suggested by the reviewer is an important control to ensure that the hae hsl2-Aeq line can respond to a Ca2+ inducing peptide signaling through a different receptor than HAE or HSL2. One would expect to see a Ca2+ response in this line to the flg22 peptide. We performed this experiment and surprisingly we could not detect a flgg22 induced Ca2+ signal in the hae hsl2 mutnt. As it is unlikely that the Ca2+ response triggered by flg22 is dependent on HAE and HSL2 we have to assume that the lack of response is due to a malfunction of the Aeq sensor in this line. As a control to measure the amount of Aeq present in the cells we treat the Aeq seedlings with 2 M CaCl2 and measure the luminescence constantly for 180 seconds (Ranf et al., 2012, DOI10.1093/mp/ssr064). The CaCl2 treatment disrupts the cells and releases the Aeq sensor into the solution where it will react with Ca2+ and release the total possible response in the sample (Lmax) in form of a luminescent peak. When treating the hae hsl2-Aeq line with CaCl2we observe a luminescent peak, indicating the presence of the sensor, however, the response is reduced compared to WT seedlings expressing Aeq. Given the sensitivity of FLS2 to flg22 one would still expect to see a Ca2+ peak in the hae hsl2-Aeq line even if the amount of sensor is reduced. Given that this is not the case, we have to assume that localization or conformation of the sensor is somehow affected in this line or that there is another biological explanation that we cannot explain at the moment.

We have therefore opted on omitting the results using the hae hsl2 Aeq lines from the manuscript and are in the process of mutating HAE and HSL2 by CRISPR-Cas9 in the Aeq background to verify that the mIDA triggered Ca2+ response is dependent on HAE and HSL2.

**Reviewer #2 (Public Review):**
Lalun and co-authors investigate the signalling outputs triggered by the perception of IDA, a plant peptide regulating organs abscission. The authors observed that IDA perception leads to a transient influx of Ca2+, to the production of reactive oxygen species in the apoplast, and to an increase accumulation of transcripts which are also responsive to an immunogenic epitope of bacterial flagellin, flg22. The authors show that IDA is transcriptionally upregulated in response to several biotic and abiotic stimuli. Finally, based on the similarities in the molecular responses triggered by IDA and elicitors (such as flg22) the authors proposed that IDA has a dual function in modulating abscission and immunity. The manuscript is rather descriptive and provide little information regarding IDA signalling per se. A potential functional link between IDA signalling and immune signalling remains speculative.

We thank the reviewer for the concerns raised and agree that further experiments including pathogen assays would strengthen the link between IDA signaling and immunity and plan for such experiments in future work.

**Reviewer #3 (Public Review):**
Previously, it has been shown the essential role of IDA peptide and HAESA receptor families in driving various cell separation processes such as abscission of flowers as a natural developmental process, of leaves as a defense mechanism when plants are under pathogenic attack or at the lateral root emergence and root tip cell sloughing. In this work, Olsson et al. show for the first time the possible role of IDA peptide in triggering plant innate immunity after the cell separation process occurred. Such an event has been previously proposed to take place in order to seal open remaining tissue after cell separation to avoid creating an entry point for opportunistic pathogens.The elegant experiments in this work demonstrate that IDA peptide is triggering the defenseassociated marker genes together with immune specific responses including release of ROS and intracellular CA2+. Thus, the work highlights an intriguing direct link between endogenous cell wall remodeling and plant immunity. Moreover, the upregulation of IDA in response to abiotic and especially biotic stimuli are providing a valuable indication for potential involvement of HAE/IDA signalling in other processes than plant development.

We are pleased that the reviewer finds our findings linking IDA to defense interesting and would like to thank the reviewer for this positive feedback.

Strengths:The various methods and different approaches chosen by the authors consolidates the additional new role for a hormone-peptide such as IDA. The involvement of IDA in triggering of the immunity complex process represents a further step in understanding what happens after cell separation occurs. The Ca2+ and ROS imaging and measurements together with using the haehsl2 and haehsl2 p35S::HAE-YFP genotypes provide a robust quantification of defense responses activation. While Ca2+ and ROS can be detected after applying the IDA treatment after the occurrence of cell separation it is adequately shown that the enzymes responsible for ROS production, RBOHD and RBOHF, are not implicated in the floral abscission.Furthermore, IDA production is triggered by biotic and abiotic factors such as flg22, a bacterial elicitor, fungi, mannitol or salt, while the mature IDA is activating the production of FRK1, MYB51 and PEP3, genes known for being part of plant defense process.

Thank you.

Weaknesses:Even though there is shown a clear involvement of IDA in activating the after-cell separation immune system, the use of p35S:HAE-YFP line represent a weak point in the scientific demonstration. The mentioned line is driving the HAE receptor by a constitutive promoter, capable of loading the plant with HAE protein without discriminating on a specific tissue. Since it is known that IDA family consist of more members distributed in various tissues, it is very difficult to fully differentiate the effects of HAE present ubiquitously.

We agree on this statement. Nevertheless, it is important to note that the responses we have observed are not detectable in WT plants that do not (over)express the HAE receptors. Suggesting that the ROS and callose deposition are induced by the addition of mIDA peptide and not the potential presence of the endogenous IDL peptides.

The co-localization of HAE/HSL2 and FLS2 receptors is a valuable point to address since in the present work, the marker lines presented do not get activated in the same cell types of the root tissues which renders the idea of nanodomains co-localization (as hypothetically written in the discussion) rather unlikely.

Thank you for raising an important aspect of our study. It is true that not all cells in the root which have promoter activity for FLS2 also exhibit promoter activity for either HAE or HSL2. However, we have observed that certain cells in the roots show promoter activity for both receptors. In the revised version of the manuscript, we have included plants expression a transcriptional promoter for both FLS2 and HAE or HSL2 using different fluorescent proteins. We have investigated overlapping promoter activity both at sites of lateral roots, in the tip of the primary root and in the abscission zone. Our results show overlapping expression of the transcriptional reporters in certain cells, indicating that FLS2 and HAE or HSL2 are likely to be found in some of the same cells during plant development. We also observe cells where only one or none of the promoters are active.

**Recommendations for the authors:**

**Reviewer #1 (Recommendations For The Authors):**
Supplementary Figure 3: re-labelling of y axis; 200 than 200,00 for clarity.

This has been addressed.

Supplementary Figure 2: It would be good to include the age of the seedlings used to study calcium influx in the legend.

This has been addressed.

Supplementary Figure 1: rephrase 'IDA induces ROS production in Arabidopsis'.

This has been addressed.

The use of chelating agents to establish the need of calcium from extracellular space is a clear experiment supporting the calcium response phenotype specific to IDA treatment in seedlings. Removing the last asparagine (N) and using it as a peptide that fails to elicit calcium response could simply be because of the peptide is smaller in length or different chemical properties. Therefore, a scrambled sequence would have been a better control.

We thank the reviewer for the suggestion of using a scrambled peptide as a negative control, however we find it unlikely that mIDA∆N69 could induce any activity based on previous work. Results from crystal structure of mIDA bound to the HAE receptor and ligand-receptor interaction studies (10.7554/eLife.15075 ) show that the last asparagine in the mIDA peptide is essential for detectable binding to the HAE receptor and that a peptide lacking this amino acid does not have any activity. We will however, in future experiments also include a scrambled version of the peptide as an additional control.

**Reviewer #2 (Recommendations For The Authors):**
Please find below specific comments:(1) Most of the molecular outputs triggered by IDA can be considered as common molecular marks of plant peptides signalling, they do not represent strong evidences of a potential function of IDA in modulating immunity. For instance, perception of CIF peptides, which control the establishment of the Casparian strips, regulate the production of reactive oxygen species, and the transcription of genes associated with immune responses (Fujita et al., The EMBO Journal 2020). It should also be considered that FRK1, whose function remains unknown, may be involved in both immunity and abscission and that the upregulation of FRK1 upon IDA treatment is not indicative of active modulation of immune signalling by IDA.

This is a fair point raised by the reviewer and we now address in the manuscript that ROS and Ca2+ are hallmarks of both plant development and defense. The function of FRK1 is not known however, it is unlikely that the upregulation of FRK1 in response to mIDA plays a role in the developmental progression of abscission as it is not temporally regulated during the abscission process, thus making it an unlikely candidate in the regulation of cell separation (Cai & Lashbrook, 2008, https://doi.org/10.1104/pp.107.110908). We do however agree that further experiments including pathogen assays would strengthen the link between IDA signaling and immunity and plan for such experiments in future work.

(2) It remains unknown whether IDA modulate immunity. For instance, does IDA perception promote resistance to bacteria (bacterial proliferation, disease symptoms)? Is IDA genetically required for plant disease resistance immunity? Is the IDA signalling pathway genetically required for transcriptional changes induced by flg22, such as increase in FRK1 transcripts? In addition, the authors propose that the proposed function of IDA in modulating immune signalling prevents bacterial infection in tissue exposed to stress(es). Does loss of function of IDA or of its corresponding receptors leads to changes in the ability of bacteria to colonise plant root upon stress(es)?

Please see the comment above regarding pathogen assays.

(3) Several aspects of the work appear to correspond to preliminary investigation. For instance, the authors analyse loss of function mutant for genes encoding for Ca2+ permeable channels (CNGCs) which are transcriptionally active during the onset of abscission (Sup. Figure 5). None of the single mutants present an abscission defect. These observations provide no information regarding the identity of the channel(s) involved in IDA-induced calcium influx.

We agree with the reviewer that we have not been able to identify the channels responsible for the IDA-induced calcium influx. Given the redundancy for many of the members of this multigenic family a future approach to identify proteins responsible for the IDA triggered calcium response could be to create multiple KO mutants by CRISPR Cas9.

(4) Using H2DCF-DA, the authors observed a decrease in ROS accumulation in the abscission zone of rbohd/rbohf double KO line (Sup Figure 5c) but describe in the text that ROS production in this zone does not depend on RBOHD and RBOHF (L220). Please clarify.

This has now been clarified in the text.

(5) The authors describe that rbohd/rbohf double KO present a lower petal break-strength, which they describe as an indication of premature cell wall loosening, and that petals of rbohd/rbohf abscised one position earlier than in WT. Yet, the authors postulate that IDA-induced ROS production does not regulate abscission but may regulate additional responses. Instead the data seems to indicate that ROS production by RBOHD and RBOHF regulate the timing of abscission. In addition, it would have been interesting to test whether IDA signalling pathway regulate ROS production in the abscission zone.

The rbohd and rbohf double mutants show several phenotypes associated to developmental stress, the mild phenotype observed with regards to premature abscission (by one position) could be caused by the phenotype of the double mutant rather than related to ROS production. Indeed, it has been suggested that the lignified brace in the AZ dependent on ROS production by the aforementioned RBOHs in necessary for the correct concentration of cell modifying enzymes (Lee et al., 2018, https://doi.org/10.1016/j.cell.2018.03.060). The precocious abscission in this double mutant clearly shows this not to be the case. We have tried to do a ROS burst assay on AZ tissue/flowers with the mIDA peptide but have not been successful with this approach. A ROS sensor expressed in AZ tissue would be a valuable tool to address whether IDA signalling regulates ROS production in AZs.

(6) In Sup. Figure5a, it would be of interest to have a direct comparison of the transcript accumulation of the presented CNGCs and RBOHDs with other of these multigenic families.

The CNGCs and RBOH gene expression profile shown in the figure are the family members expressed during the developmental progress of floral abscission in stamen AZs. Since there is no difference in the temporal expression of the other family members (and most are either not expressed or very weakly expressed in this tissue) it is not possible to do this comparison (Cai & Lashbrook, 2008, https://doi.org/10.1104/pp.107.110908).

(7) L251-253, since IDAdeltaN69 cannot be perceived by its receptors, the absence of induction of pIDA::GUS by IDAdeltaN69 compared to flg22 cannot be seen as a sign of specificity in peptideinduced increase in IDA promotor activity.

We have rephased this in the text

(8) Please provide quantitative and statistical analysis of the calcium measurement presented in sup figure 3.

This has been addressed.

(9) L339-341; This sentence is unclear to me, please rephrase.

We have rephased this in the text

**Reviewer #3 (Recommendations For The Authors):**
(1) In order to assess the role of CNGCs in abscission process, it would be more interesting to see the effect on the Ca2+ pattern and ROS signaling after application of mIDA on cngc and rbohf rbohd mutants.

We agree in this statement and the studies on mIDA induced ROS and Ca2+ on these mutants will provide valuable information to the regulation of the response. We are in the process of making the lines needed to be able to perform these experiments. However, since it requires crossing of genetically encoded sensors into each mutant, and generation of higher order mutants this is a long process.

(2) With regard to the ROS production (Sup Fig. 1), the application of mIDA can trigger ROS in p35S::HAE:YFP lines, but not in the wild-type plant, which is according to the text "most likely due to the absence of HAE expression" in leaves. The experiment on callose deposition is performed in wild-type cotyledons where no callose deposition could be observed after mIDA treatment (Fig. 4a,b). The conclusion from text is that IDA "is not involved in promoting deposition of callose as a long-term defence response". It appears more likely that neither ROS nor callose can be observed in wild-type plants due to the lack of HAE expression. Therefore, the callose experiment should include the p35S::HAE:YFP lines. The experiment as it is does not allow to draw any conclusion on HAE/IDA involvement in callose formation.

We fully agree with this comment, thank you for pinpointing this out. We have now performed the callose experiment with the 35S:HAE lines. Please see our answer to reviewer #1.

(3) Between Sup Fig. 3 and Sup Fig. 5 two different systems were used to asses the floral stage. An adjustment of the floral stages would be easier to convey the levels of HAE/HSL2 expression and hence potentially with the onset of cell-wall degradation.

We now used the same system to assess floral stages throughout the whole manuscript.

(4) For the Fig. 1 and 2, it will be helpful to mention the genotype used for imaging/quantification of Ca2+.

This has been addressed.

(5) Some of the abbreviations are not introduced as full-text at their first time use in the text, such as: mIDA (Line 68), Ef-Tu (line 85), NADPH (line 77).

The abbreviations have now been introduced.

(6) In the legend of Fig. 5 (lines 897 and 898)- in the figure description, the box plots are identified as light gray and dark gray, while in the panel a of the figure the box plots are colored in red and blue.

Thank you for pointing this out, this has now been corrected.

(7) In figure 1 and 2. the authors write that the number of replicates is 10 (n=10) but data represents a single analysis. Please provide the quantitative ROI analysis, demonstrating that the observed example is representative. This is particularly important since the authors claim very specific changes in pattern of Ca signaling between mIDA and FLG22 treatments (Line 148).(8) Figure 4: please use alternative scaling on the Y axis instead of breaks.

This has now been fixed.

(9) Figure 5: it is not clear what n=4 refers to when the authors state three independent replicates. In figure 6 they state 4 technical reps and 3 biological reps. Please ensure this is similar across all descriptions.

We have now ensured the correct information in all descriptions.